# EPISODIC MEMORY THEORY FOR THE MECHANISTIC INTERPRETATION OF RECURRENT NEURAL NETWORKS

## ABSTRACT

Understanding the intricate operations of Recurrent Neural Networks (RNNs) mechanistically is pivotal for advancing their capabilities and applications. In this pursuit, we propose the Episodic Memory Theory (EMT), illustrating that RNNs can be conceptualized as discrete-time analogs of the recently proposed General Sequential Episodic Memory Model. To substantiate EMT, we introduce a novel set of algorithmic tasks tailored to probe the variable binding behavior in RNNs. Utilizing the EMT, we formulate a mathematically rigorous circuit that facilitates variable binding in these tasks. Our empirical investigations reveal that trained RNNs consistently converge to the variable binding circuit, indicating universality in the learned dynamics of RNNs. Building on these findings, we devise an algorithm to define a *privileged basis*, which reveals latent neurons instrumental in the temporal storage and composition of variables — a mechanism vital for the successful generalization in these tasks. We show that the privileged basis enhances the interpretability of the learned parameters and hidden states of RNNs. Our work represents a step toward demystifying the internal mechanisms of RNNs and, for computational neuroscience, serves to bridge the gap between artificial neural networks and neural memory models.

## 1 INTRODUCTION

AI-driven systems have become ubiquitous in real-world applications (Christian, 2021; Sears, 2021; Bostrom, 2014; Müller & Bostrom, 2013). While these systems demonstrate remarkable proficiency, their inherently black-box nature often renders them inscrutable (Alishahi et al., 2019; Buhrmester et al., 2019; Fong & Vedaldi, 2017). Mechanistic interpretability aims to reverse engineer the intricate workings of neural networks that drive their behavior (Olah, 2022). Gaining a mechanistic understanding builds trust in AI systems and provides insights that can lead to refinement and innovation (Raukur et al., 2022). In essence, mechanistic interpretability is not just about demystifying AI; it's about harnessing its potential responsibly and efficiently. Recurrent Neural Networks (RNNs) (Hochreiter & Schmidhuber, 1997) play a pivotal role in AI due to their unique ability to process sequential data (Graves et al., 2013), making them indispensable for tasks involving time series analysis, natural language processing, and other applications where understanding temporal dynamics is crucial (Che et al., 2018). One major challenge in understanding RNNs mechanistically is that the task-relevant information is stored in a hidden state that evolves over time. This temporal nature of RNNs raises critical questions: *How is information reliably stored and processed in this evolving hidden state?* and *How are the learned parameters of RNNs connected to the computations performed?* Addressing these questions is vital for advancing our understanding and application of RNNs in AI-driven systems.

Answering these questions in RNNs require elucidating the mechanisms of 'variable binding' that enables them to dynamically associate information with variables and manipulate the information symbolically over time (Marcus, 2001). In cognitive systems, variable binding enables generalization in complex, structured tasks that involve symbolic relationships and dependencies between various elements (Greff et al., 2020). For instance, in natural language processing, variable binding promotes understanding and maintaining context over a sentence or a conversation. The importance of uncovering the variable binding mechanisms stems from its potential to bridge the gap between

simple pattern recognition and advanced cognitive capabilities, and move towards a better understanding and reasoning of AI systems. This will not only enhance the capabilities of AI systems but also provides deeper insights into the nature of intelligence itself - both artificial and biological.

**Organization**: To formulate variable binding mechanisms in RNNs, we turn to computational neuroscience, drawing parallels between autonomously evolving RNNs and episodic memory retrieval models. First, we show the connection between RNN architectures and a recently proposed episodic memory model - General Sequential Episodic Memory Model in Section 3. We show that while GSEMM was introduced in continuous time, its temporal discretization corresponds with the evolution of RNNs. Episodic memory has varied definitions in different fields. Our definition of episodic memory is in line with the General Sequential Episodic Memory Model (Karuvally et al., 2022) and neuroscience (Umbach et al., 2020), which describes the ability of neural networks to store and process temporal and contiguous memory sequences. This contrasts with the psychological perspective of episodic memory as the subjective recollection of personal experiences (Tulving, 1972) where the focus is on the human recollecting the memory rather than the underlying system.

In Section 4, we develop a class of algorithmic tasks designed to investigate the variable binding mechanisms of RNNs. These tasks involve a two-phase process: in the input phase, an RNN is presented with a series of $d$-dimensional vectors over $s$ timesteps, and in the output phase, it autonomously generates outputs based on this stored information, using a linear binary symbolic composition function. The tasks, while simpler than complex real-world scenarios, builds upon and extends previous task setups that explore RNN memory capabilities (Graves et al., 2014). Section 5 introduces the concept of *variable memories*—linear subspaces within the RNN that facilitate the variable binding and recursive composition of information. This concept allows us to *fully* deconstruct the mechanisms of variable binding in RNNs and propose a circuit mechanism answering how RNNs store and process information over time. Our experimental findings demonstrate a consistent convergence to the proposed circuit, contributing evidence to the 'universality hypothesis' in mechanistic interpretability (Olah et al., 2020; Li et al., 2015a). Further, the circuit mechanisms we found show notable similarities to recently developed brain-inspired traveling waves in RNNs (Keller et al., 2023), indicating a broader applicability of the theory beyond the toy variable binding tasks.

In Section 6, we leverage the empirical convergence result to propose an algorithm to construct a privileged basis of the *variable memories*. In our results, we show that this basis fully deconstructs the learned behavior by uncovering *latent neurons* (by basis change of the RNN hidden state) and *latent synaptic interactions* (by basis change of the learned interactions) involved in information processing.

## 2 RELATED WORKS

Our exploration of RNNs spans three, often separate, research direction - Dynamical Systems interpretation of RNNs, Mechanistic Interpretability, and Neural Memory Models.

**Dynamical Systems Interpretation of RNNs**: Current approaches to interpret RNNs consider them as non-linear dynamical systems and apply linearization around fixed or slow-changing points to reveal their behavior (Marschall & Savin, 2023; Sussillo & Barak, 2013). The preliminary step in this analysis involves linearization around fixed points and slow-changing points found using optimization algorithms. The phase space flow is assembled piecemeal from each linearized region. The exploration of the long-term behavior of these regions is undertaken through the eigen-spectrum analysis of the corresponding linearized dynamical systems (Strogatz, 1994), providing insights into the dynamics of convergence, divergence, stability, or spiraling (Rowley et al., 2009; Kim, 1996). However, this method becomes intractable in our variable binding tasks when there are many dimensions exhibiting non-convergent behaviors. The proposed EMT generalizes this approach to the class of variable binding tasks and enables interpretation even when the number of non-converging dimensions is arbitrarily large (Appendix Figure 5).

**Mechanistic Interpretability**: Mechanistic interpretability seeks to reverse-engineer neural networks to expose the underlying mechanisms enabling them to learn and adapt to previously unencountered conditions. The prevailing strategy involves examining the networks' internal "circuits" (Conmy et al., 2023; Wang et al., 2022; Cammarata et al., 2020). Researchers have found that apply-

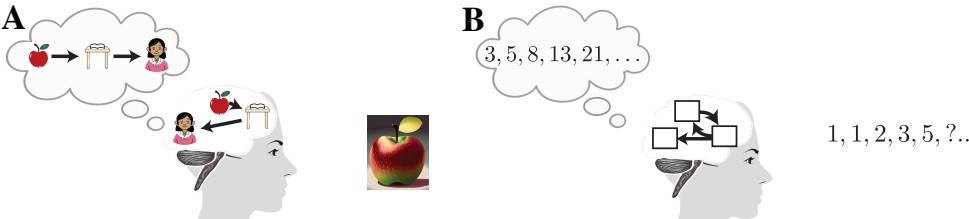

Figure 1: **Equivalence between Episodic Memory Models and Variable Binding**: **A**. Episodic Memory models aim to uncover the cognitive processes involved in the retrieval of subjective past experiences often stored as a temporal sequence of memory items. The illustration shows the retrieval of a personal experience when an apple is observed. **B**. The illustration shows the application of the Episodic Memory Theory, which poses that learning the addition operation over arbitrary numbers to generate fibonacci numbers, a task involving variable binding, can be considered equivalent to episodic memory retrieval where the computations are performed over variables instead of predetermined memories. The abstract addition operation is stored in the synapses in the form of *how* the variables interact with each other to produce the desired result (See Appendix A.3 for details).

ing interpretability methods to large networks, such as transformers (Vaswani et al., 2017) handling complex tasks in natural language processing and vision, faces the challenge of unclear features to be modeled in internal circuits. To address this challenge, toy models are created with clearly defined features essential for task resolution. Probing models trained on toy tasks has resulted in supporting evidence for prevalent hypotheses. Some of the notable hypotheses are *universality* (Chughtai et al., 2023; Li et al., 2015b) - models learn similar features and circuits across different conditions when trained on similar tasks, *bottleneck superposition* (Elhage et al., 2022) - a mechanism for storing more information than the available dimensions, and *composable linear representations* (Cheung et al., 2019) - the use of linear spaces in feature representation. Despite these advancements, current approaches remain confined to forward only models like MLPs and transformers. Our proposed EMT generalizes the circuit approach of mechanistic interpretability to recurrent architectures and provides a mathematically grounded framework for deconstructing their behavior.

**Neural Memory Models**: Developments in memory modeling have revealed links between deep neural networks and memory models. The first investigation of this link explored how the different activation functions in Multi-Layer Perceptrons affected the learned representations and memory capacity (Krotov & Hopfield, 2016). Later studies extended this connection to explain the practical computational benefits observed in neural architectures like transformers (Ramsauer et al., 2020). Recently, the traditional memory models capable of associative recall of static memories were expanded to retrieving memory sequences (Karuvally et al., 2022; Chaudhry et al., 2023). This expansion allows memories that previously did not interact in the static memory retrieval context to interact and produce complex temporal behavior (Kleinfeld, 1986; Kleinfeld & Sompolinsky, 1988). A fundamental assumption in memory modeling (in both static and sequence retrieval) is that the network's memories are predetermined and stored in the synapses. This assumption limits the models' applicability to understanding symbolic binding of memories typically available only during inference. In EMT, we will demonstrate that by lifting the fixed memory assumption in memory modeling, these memory models can be utilized to build principled circuits to show how RNNs bind external information.

**Summary**: EMT reveals the synergistic relationship between the three fields - dynamical systems interpretation of RNNs, mechanistic interpretability, and neural memory modeling and suggests a unified approach to address the challenges of understanding neural behavior.

## 3  RNN AS EPISODIC MEMORY

We show that RNNs can be viewed as a discrete-time analog of a memory model called General Sequential Episodic Memory Model (GSEMM) (Karuvally et al., 2022), and as a result, enable human-interpretability in terms of learned memories and their interaction. See Appendix A.2 for detailed proof. The sketch of the proof is detailed below. To be applicable for the more gen-

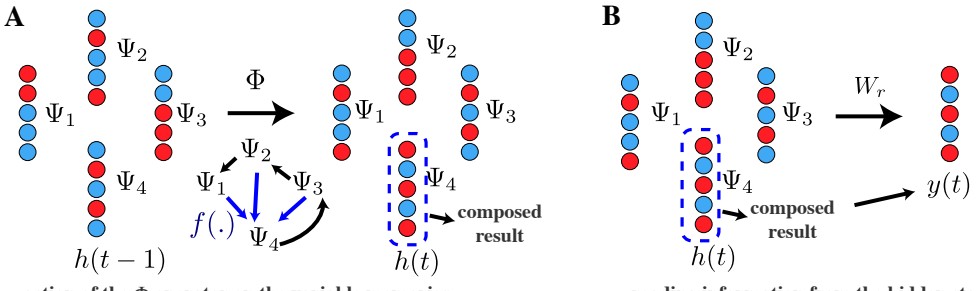

Figure 2: **Circuit of variable binding in an illustrative task of four variables, each with five dimensions**: **A**. The hidden state at time $t$ has subspaces capable of storing external information in their activities. The colors depict the vector components (or activity) of the hidden state in the variable memory basis. The linear operator $\Phi$ acts on the hidden state such that these activities are copied between variable memories except for $\Psi_4$, which implements the linear operation $f$. **B**. The $N^{\text{th}}$ variable contents are read during the output phase using the appropriate linear operator $W_r = \Psi_N^*$.

eral setting of RNNs, we slightly modify the GSEMM synapse formulation to a pseudoinverse learning rule instead of the previous Hebbian rule. This modification allows the model to handle the the more general case of linearly independent memory vectors, instead of orthogonal vectors only (Chaudhry et al., 2023; Personnaz et al., 1986). GSEMM is still a continous time memory, so we discretize it using the forward Euler approximation under the conditions that the timescale parameters of GSEMM are $\mathcal{T}_f = 1, \mathcal{T}_h = 0$, and $\mathcal{T}_d = 0$. The final discrete system we obtain is $V_f(t+1) = \Xi\,(I + \Phi'^{\top})\,\Xi^{\dagger}\,\sigma_f(V_f(t))$, where $V_f$ is a vector representing the state of the neural network, $\Xi$ and $\Phi$ are matrices representing the stored memories and inter-memory interactions respectively. The columns of the $\Xi$ matrix are the *static stored memories*, and the matrix $(I + \Phi'^{\top}) = \Phi^{\top}$ directs the chronology of these memories to form the sequences. The discrete system we derived is topologically conjugate to the update equations of an Elman RNN under the homeomorphic transformation $V_f = $ if the norm of the matrix is bounded by 1. That is, if $||\Xi\,\Phi^{\top}\,\Xi^{\dagger}|| \le 1$, we can consider a new state variable $h = \sigma_f(V_f)$ such that

$$h(t) = \sigma_f(\Xi\,\Phi^{\top}\,\Xi^{\dagger}h(t-1))\,. \tag{1}$$

This conjugate system has equations that are equivalent to an Elman RNN hidden state update equation without bias $h(t + 1) = \sigma_f(W_{hh}h(t))$.

**Corollary 3.0.1** *The learned synaptic interactions of autonomously evolving Elman RNNs without bias can be decomposed in terms of memory models ($W_{hh} = \Xi\Phi\Xi^{\top}$) and interpreted as the retrieval of memories temporally transitioning according to the rules encoded in the intermemory interactions.*

This corollary also generalizes to the case of forward-only networks as they can be viewed as a single-step update of the RNN update equations. We now formulate the Episodic Memory Theory as follows: **The Episodic Memory Theory (EMT) poses that the inner workings of learned neural networks can be revealed by analyzing the learned inter-memory interactions and its effect in the space of stored memories** (Figure 1).

## 4 VARIABLE BINDING TASKS

**Definition of Variable Binding**: *Variable binding in the context of RNNs refers to the network's ability to store and process pieces of input information symbolically across different timesteps, utilizing this information to generate the necessary outputs.*

For example, in a language translation task, variable binding involves storing the source sentence provided to the RNN as input in the hidden state which will be referred to when generating the translated language. Directly analyzing the variable binding behavior of RNNs in complex tasks like language translation is very challenging because it is not clear what the variables will be. We thus take

---

**Algorithm 1** Algorithm for approximating variable memories of trained linear RNNs

---

$0 \leq \alpha \leq 1$
$s$                                    $\triangleright$ number of time-steps in the input phase
$W_{hh}, W_r$                           $\triangleright$ learned parameters of the RNN
$\Psi_s \leftarrow W_r^\dagger$
**for** $k \in \{s-1, s-2, \dots 1\}$ **do**
    $\Psi_k \leftarrow \left( \left( W_{hh}^\top \right)^k W_r^\dagger \right)$
    $\Psi_k \leftarrow \Psi_k - EE^\dagger \Psi_k \quad \forall E : \lambda(E) < 1 \triangleright$ Remove the components along transient directions.
**end for**
$\Psi \leftarrow [\Psi_1; \dots ; \Psi_s]$
$\Psi^\perp \leftarrow \text{PC}(\{\tilde{h(t)}\} - \Psi\Psi^\dagger \{\tilde{h}(t)\})$            $\triangleright$ Principle Components of $\tilde{h}$ from simulations

---

the approach of developing a class of toy variable binding tasks where the variables that need to be stored are well-understood. This approach is used in the mechanistic interpretation of forward-only neural networks (Chughtai et al., 2023; Li et al., 2015b), and RNNs (Maheswaranathan et al., 2019; Graves et al., 2014; Sussillo & Barak, 2013). The variable binding tasks we consider are the generalization of the RepeatCopy task used to evaluate memory storage capabilities of recurrent neural architectures used by Keller et al. (2023) and Graves et al. (2014). Our approach to interpreting the trained RNNs also generalizes the dynamical systems approach of (Sussillo & Barak, 2013) to high dimensional task spaces found in the variable binding tasks (See Appendix Figure 5 for the diversity and high dimensional nature of the eigenvalue distribution found in the learned representation of the variable binding tasks). By taking the step to generalize existing simple setups, the variable binding tasks provide a path forward to close the gap between simple tasks and real-world scenarios, without sacrificing on human-interpretability.

**Variable Binding Tasks:** We define variable binding tasks as consisting of two phases: the input phase and the output phase. The input phase lasts for $s$ timesteps. During each timestep $t$ (where $1 \leq t \leq s$) of this phase, the RNN receives a $d$-dimensional input vector $u(t) = [u^1(t), u^2(t), \dots, u^d(t)]^\top$. These vectors $u(t)$ provide the external information that the RNN is expected to store within its hidden state. It is important to note that the information necessary to compute the output at each timestep is cumulatively stored for use in subsequent steps.

Once the input phase concludes at timestep $s$, the output phase begins immediately from timestep $s + 1$. During the output phase, the RNN no longer receives external input and instead operates autonomously, generating outputs based on the information stored during the input phase. The RNN should output the composition function

$$y(t) = f(y(t-1), y(t-2), \dots y(t-s)), \text{ for } t > s \tag{2}$$

$y(t)$ is initialized as $y(t) = u(t)$ for $0 < t \leq s$. This task setup implies that the RNN needs to use the accumulated information from the input phase to influence its outputs in the subsequent phase.

For the purpose of our analysis in the paper, we define two constraints on the variable binding tasks: (1) the composition function $f$, which governs the output generation, is linear, and (2) the domain of $f$ (the set of possible input values), and the codomain of $f$ (the set of possible output values) are binary, consisting of values in $\{-1, 1\}$. These simplifications enable us to focus on *how* the RNN processes and integrate the input information across different timesteps to produce coherent outputs, abstracting out *what* the variables are actually storing.

## 5 VARIABLE BINDING CIRCUIT IN RNN

In this section, we develop a circuit to explain the mechanisms in RNNs that enable them to learn and generalize in the variable binding tasks. The circuit simplifies understanding the complex dynamics of these networks in a more analytically tractable, and human interpretable form. Our approach starts by considering RNNs in their linearized form, defined by specific linear equations involving the hidden state, input, and output. This simplification lays the foundation for further analysis. We propose changing the basis of the RNN's representation (both the hidden state and the learned synaptic interactions), to treat the linearized RNN as transitioning according to the sequential mem-

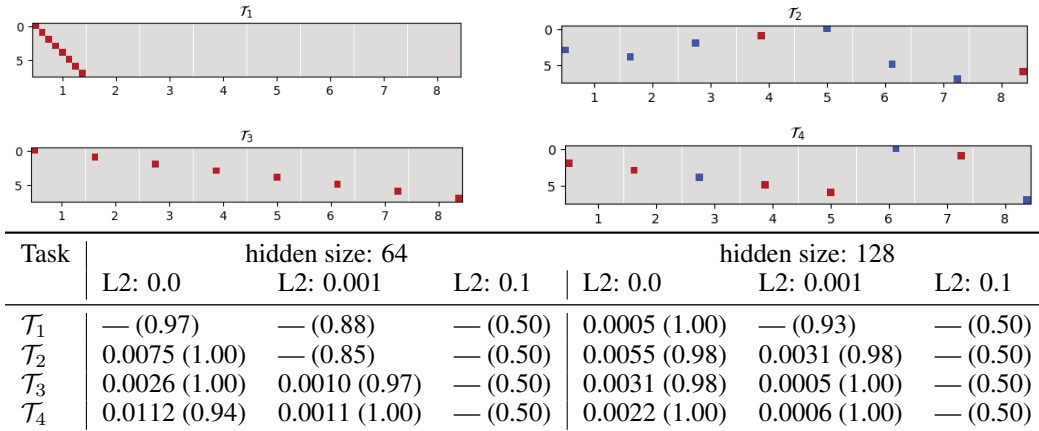

| Task | hidden size: 64 | | | hidden size: 128 | | |
|---|---|---|---|---|---|---|
| | L2: 0.0 | L2: 0.001 | L2: 0.1 | L2: 0.0 | L2: 0.001 | L2: 0.1 |
| $\mathcal{T}_1$ | — (0.97) | — (0.88) | — (0.50) | 0.0005 (1.00) | — (0.93) | — (0.50) |
| $\mathcal{T}_2$ | 0.0075 (1.00) | — (0.85) | — (0.50) | 0.0055 (0.98) | 0.0031 (0.98) | — (0.50) |
| $\mathcal{T}_3$ | 0.0026 (1.00) | 0.0010 (0.97) | — (0.50) | 0.0031 (0.98) | 0.0005 (1.00) | — (0.50) |
| $\mathcal{T}_4$ | 0.0112 (0.94) | 0.0011 (1.00) | — (0.50) | 0.0022 (1.00) | 0.0006 (1.00) | — (0.50) |

Table 1: **RNNs consistently converge to the variable binding circuit**: The top image shows the composition functions for the 4 tasks, visualized as a matrix with x-axis input, and y-axis output. Red color denotes +1, blue is -1 and no color is 0. $\mathcal{T}_1$ is the function for a simple repeat copy task, the rest are other general composition functions. The table shows the MAE in the complex argument between the eigenspectrum of the predicted $\Phi$ from the variable binding circuit and the empirically learned $W_{hh}$ in 4 tasks across 20 seeds under different RNN configurations. This average error is indeterminate (—) if the rank of the theoretical $\Phi$ is different from the empirical $W_{hh}$. Values in the brackets show the average test accuracy of the trained model. For models that have high test accuracy ($> 0.94$), the error in the theoretically predicted spectrum is very low indicating consistent convergence to the theoretical circuit. A notable exception of this behavior is $\mathcal{T}_1$ with hidden size= 64 and $L2 = 0$, where the restricted availability of dimensions forces the network to encode variables in bottleneck superposition resulting in a low-rank representation of the solution.

ory relationships of GSEMM. This basis change is essential for simplifying our understanding and deconstructing the internal computations of the RNN's dynamics. We introduce the concept of variable memory to collectively reason about the activities within subspaces of the RNN hidden state and show that these subspaces can act as variables in the RNN computation. This concept of variable memories is *not* restricted to the variable binding tasks and can be used by researchers to further develop principled models of neural computation. We finally use the concept of variable memories to propose a circuit and form principled hypotheses of what each parameter of the learned RNN is and the role they play in RNNs trained on the variable binding tasks. We later validate these hypotheses using experiments.

**Linear RNN**: We build our model of variable binding on a linearized RNN defined by the following equations.

$$\begin{cases} h(t) = W_{hh}h(t-1) + W_{uh}u(t)\,, \\ y(t) = W_r\,h(t)\,. \end{cases} \tag{3}$$

We envision that any non-linear RNN can be converted to this form by finding fixed points and subsequent linearization (See Appendix A.5 for details). Here, $W_{hh}, W_{uh}, W_r$ are linear operators that gets learned after training on the variable binding tasks. We will use the concept of variable memories to form principled hypotheses for these operators that will be validated by experimental results. $h(t)$ is the hidden state, $u(t)$ is the input, and $y(t)$ is the output. We use a simplifying assumption that $W_{hh}$ has sufficient capacity to represent all the variables required for the variable binding tasks. We further assume that $h(0)$ is the zero vector.

**Variable Memory**: We decompose the linear RNN using the GSEMM equivalence (Corollary 3.0.1) and define variable memories as subspaces of the space spanned by: $\psi_\mu = \sum_i \xi_\mu^i e_i$. In the new basis, the hidden state vector is $h(t) = \sum_\mu h^{\psi_\mu}\psi_\mu$. The subspace spanned by the collection of vectors $\{\Psi_k\} = \{\psi_\mu : \mu \in \{(k-1)d, \ldots, kd\}\}$ is called the $k^{\text{th}}$ *variable memory*. The activity of the subspace is the contents of the $k^{\text{th}}$ variable.

**Variable Memory Interactions (*hypothesis*)**: The variable binding tasks require mechanisms in $\Phi$ capable of retaining variables of the variable binding tasks for $s$ timesteps. One possibility for this

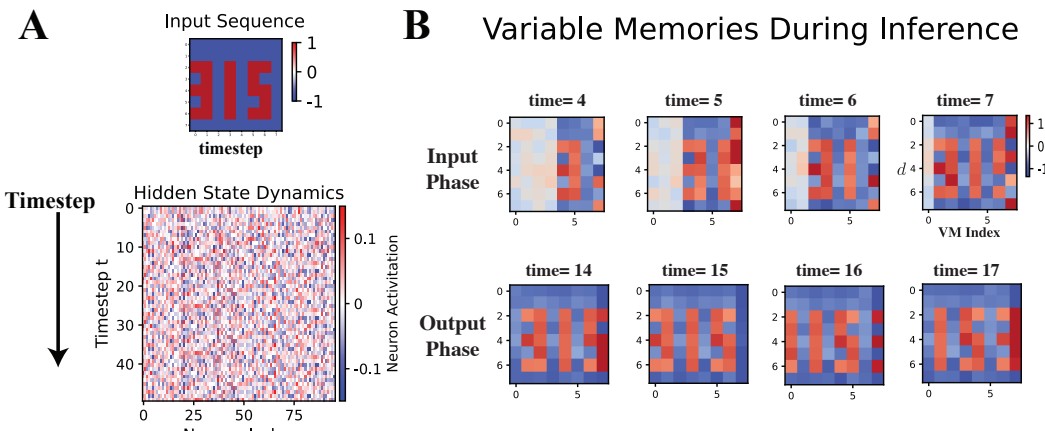

Figure 3: **EMT reveals latent neurons storing task relevant information over time**: **A**. In the repeat copy task ($\mathcal{T}_1$), the RNN needs to repeatedly produce an input sequence that is presented. A typical trained hidden state after providing the input does not show any meaningful patterns connected to the input. **B**. The same hidden states when visualized in the variable memories reveal the input information being stored as variables and processed according to the variable binding circuit. The actual hidden state is in a superposition of these *latent* variable memory activities.

is the following linear operator $\Phi$, defined in terms of the variable memories.

$$\Phi = \sum_{\mu=1}^{(s-1)d} \psi_\mu \psi^{\mu+d} + \underbrace{\sum_{\mu=(s-1)d}^{sd} \sum_{\nu=1}^{sd} \Phi_\nu^\mu \psi_\mu \psi^\nu}_{f(u(t-1),u(t-2),\dots,u(t-s))} \quad . \tag{4}$$

The action of the operator on the hidden state is illustrated in Figure 2A. For all variable memories with index $i \in \{2, 3, 4, \dots s\}$, the variable contents (subspace activities) gets copied to the variable memory with index $i - 1$. The operator then applies the function $f$ defined in Equation 2 on the variable contents and stores the result in the $s^{\text{th}}$ subspace. This circuit generalizes to any instantiation of the variable binding tasks with a *linear* composition function $f$.

**Reading from Variable Memories (*hypothesis*)**: Once RNN has performed its computation in the space of variable memories, the computed information needs to be extracted from the hidden state. The linear RNN has an operator $W_r$, which facilitates the reading of information from $h(t)$ at consecutive time steps. We propose that $W_r$ has the following equation which projects the activity of the $s^{th}$ subspace to the standard basis (Figure 2B) for output.

$$W_r = \sum_{\mu=(s-1)d+1}^{sd} e_{\mu-(s-1)d} \psi^\mu \tag{5}$$

### 5.1 RESULT: RNNS CONSISTENTLY CONVERGE TO THE VARIABLE BINDING CIRCUIT

To substantiate that the current hypothetical circuit is learned by empirical RNNs when trained on the variable binding tasks, we trained various RNN configurations, differing in hidden sizes and regularization penalties on 4 variable binding tasks each differing in the linear composition function $f$ (See top of Table 1). After training, the RNNs were linearized, and the eigen-spectrum of the learned $W_{hh}$ matrix is compared with the theoretical $\Phi$, as defined in Equation 4. If RNNs learn a representation in alignment with our model, both operators, i.e., the learned $W_{hh}$ and theoretical $\Phi$, are expected to share a portion of their eigenspectrums as they are similar matrices (i.e they differ only by a basis change). We compared only the complex arguments of the spectrum, disregarding the magnitude. The rationale behind this exclusion lies in what the magnitude tells about the dynamical behavior. The eigenvalue magnitude portrays whether a linear dynamical system is diverging, converging, or maintaining consistency along the eigenvector directions (Strogatz, 1994). RNNs typically incorporate a squashing non-linearity, such as the Tanh activation function, which restricts

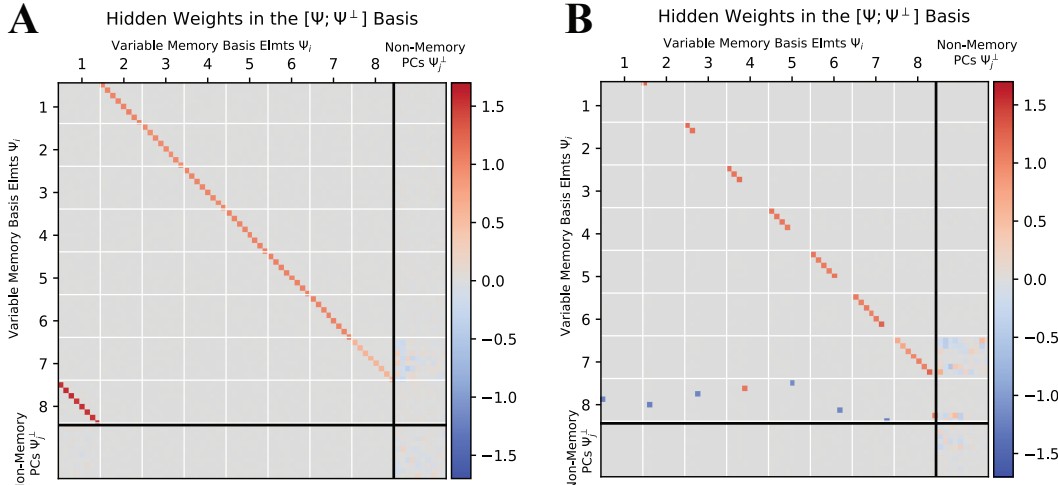

Figure 4: **EMT enables human interpretability of learned RNN parameters**: The learned weights when visualized in the variable memories result in a form that is human-interpretable. For RNNs trained on two sample tasks $\mathcal{T}_1$ (**A** left) and $\mathcal{T}_2$ (**B** right), the weight matrix $W_{hh}$ converts into a form that reveals internal mechanisms of how RNNs solve the task. For both tasks, the variables with index $< 8$ copies its contents to the preceding variable. Variable 8 actively computes the function $f$ applied on all the variables stored in the hidden state using the variable binding circuit. For $\mathcal{T}_1$, it is a simple copy of the 1st variable, and for $\mathcal{T}_2$, it is a linear composition of all the variables Notably, the circuit for $\mathcal{T}_2$ shows an optimized basis where all the irrelevant dimensions are absent.

trajectories that diverge to infinity. Essentially, provided the eigenvalue magnitude remains $\geq 1$, the complex argument solely determines the overall dynamical behavior of the RNN. Table 1 depicts the average absolute error in the eigenspectrum and test accuracy when the RNN models are trained across 4 distinct variable binding tasks. The table shows that RNNs consistently converge to the hypothetical circuit.

## 6 PRACTICAL CONSIDERATIONS OF EMT

Now we have sufficient evidence to show that the learned parameters of RNNs converge to our hypothetical circuit of Equation 4. We can now develop a procedure to explore empirical RNNs in terms of the variable memories. The RNN can then be interpreted using the variable binding circuit to deconstruct *where* the variable memories are stored and *how* they interact to produce the necessary outputs. Viewing the operations of the learned RNN interaction matrix $W_{hh}$ in the basis of variable memories has an additional benefit – it provides a path to influence or "fix" RNN behavior after the training. This "fixing" operation can be imagined as changing the learned parameter $W_{hh}$ by influencing specific weights of the extracted $\Phi$. This can potentially be utilized to improve variable storage characteristics, fix problems computing the composition $f$ or even remove computations that are deemed unnecessary.

Building on the intuition from the linear model, we use the learned RNN weights $W_{hh}$, and $W_r$ to estimate the basis vectors of the variable memories. From our hypothesis on reading from variable memories (Equation 5), $\Psi_s = W_r^\dagger$ († is the Moore-Penrose pseudoinverse) defines a matrix whose columns are the basis vectors of the $s$th variable memory. Similarly, other basis vectors can be found as columns of the matrices obtained by propagating these dimensions *forward* in time: $\Psi_k = \Phi^{s-k}\Psi_s = W_{hh}^{s-k}W_r^\dagger$. Although the variable memories are defined based on a linear evolution assumption, we found that the method of power iterating $W_{hh}$ was effective in defining the variable memories for even the non-linear Elman RNNs. For completeness, we characterize RNN behavior not currently explainable by the variable binding circuit using a space orthogonal to the variable memories ($\Psi^\perp$). The pseudo-code for the algorithm to approximate variable memories from a linearized RNN is summarized in Algorithm 1. A formal analysis of the correctness of the approximation is in the Appendix A.6.

## 6.1 RESULT: VARIABLE MEMORY REVEALS VARIABLE BINDING LATENT NEURONS

We approximated the variable memories of RNNs trained on the Repeat Copy task ($\mathcal{T}_1$) using the algorithm and visualized the hidden state. In the Repeat Copy task, the RNN must repeatedly output the stream of inputs provided during the input phase. The simulated hidden states of learned RNNs are visualized by projecting the hidden state in the variable memories: $\tilde{h} = \Psi\Psi^\dagger h$. The results shown in Figure 3 reveal that the hidden state is in a superposition (or distributed representation) of latent neurons that actively store each variable required to compute the function $f$ at all points in time. The basis transformation helps to disentangle these superposed (or distributed) variables from the hidden state so that they are easily visualized.

## 6.2 RESULT: VARIABLE MEMORIES ENABLE HUMAN INTERPRETABILITY OF LEARNED WEIGHTS

In addition to revealing hidden neurons that store and process information over time, variable memories can also be used as bases to view the function of the learned matrices. The variable memories are carefully constructed such that $W_{hh}$ converts to the underlying $\Phi$ when viewed in the basis and any behavior not explainable by $\Phi$ is shown as interactions with the orthogonal space. The Figure 4 shows the learned parameters of $W_{hh}$ encoding the variable binding circuit. The low connectivity (near zero magnitude in the interactions) between the variable memories and the orthogonal space indicates that the variable binding circuit fully explains the behavior of the RNNs.

## 7 DISCUSSION

In this work, we frame Recurrent Neural Networks as discrete-time analogs of the General Sequential Episodic Memory Model – an energy-based memory model from computational neuroscience. We introduced the concept of "variable memories," linear subspaces capable of symbolically binding and recursively composing information, providing a path forward to lift the fixed memory assumption of the memory model, and promoting applicability in mechanistic understanding. The variable memory approach addresses some of the limitations of current methods in understanding RNNs, particularly the intractability of Jacobian spectral analysis in high-dimensional task spaces. We presented a new class of algorithmic tasks that are designed to probe the variable binding behavior by generalizing existing simple tasks, taking a step to close the gap between toy tasks and real-world tasks without compromising on human interpretability. We presented a circuit mechanism that is capable of recursively storing and composing variables and showed empirical evidence that trained RNNs consistently converge to this circuit indicating computational universality in the learned representation and behavior of models. Building on the evidence, we used variable memories to define a privileged basis from trained RNN parameters that revealed latent neurons actively involved in information processing. Further, using variable memories, we viewed the learned parameters of an RNN in a human-interpretable manner, enabling reasoning and understanding RNN behavior as repeatedly shifting and composing variables. Using the tools from the theory, we *fully* deconstructed both the hidden state behavior *and* the learned parameters of empirically trained RNNs.

**Limitations**: With these results, it is also important to recognize inherent limitations to the variable memory approach. One of the limitations is that the analysis we presented is primarily restricted to linear dynamical systems. Although an accurate representation of the qualitative behavior within *small neighborhoods* of fixed points can be found for non-linear dynamical systems, the RNNs have to be confined to these linear regions for the approach to be applicable. It is still an interesting behavior that models consistently converge to this linearized regime, at least for the tasks outlined in Section 4. Outside the linear regions, RNNs may also exhibit behaviors like ergodicity and chaos which the current analysis cannot support. The second limitation of the approach is that external information is stored as a *linear* superposition of variable memories in the hidden state. Our results indicates that the role of non-linearity in encoding external information may be minimal for the toy tasks. However, we have observed that when the number of dimensions of the linear operator $W_{hh}$ is not substantially large compared to the task's dimensionality requirements (bottleneck superposition) or when the regularization penalty is high, the RNN can effectively resort to non-linear encoding mechanisms to store external information (Appendix B.3). Overcoming these limitations of non-linearity will be an interesting direction to pursue in future research, and will bring the con-

cept of variable memories closer to addressing the challenges posed by the quest to understand neural computation.

## 8 REPRODUCIBILITY STATEMENT

We have taken steps to ensure the reproducibility of the theoretical and empirical contributions of the paper. The main contributions of the paper include: (1) The theoretical connection between RNNs and sequence memory models, (2) creation of toy models to test variable binding in RNNs, (3) a mathematical model of variable binding, and (4) algorithm to compute variable memories for RNN interpretability. For (1), we summarize the high level ideas of the proof in the main document in Section 3 and provide detailed steps in Appendix A.2. For (2), we provided a mathematical description in Section 4 where the final paragraph details the assumptions we consider in the paper. For (3), we detail the mathematical model in Section 5, the assumptions of linearity and sufficient rank is stated in the section. Further, we have provided two fully worked examples in Appendix A.4. For the theoretical component of (4), we have stated the algorithm in Algorithm 1, and explained how the algorithm was formulated in Section 6. For the empirical component of (4), we have detailed the empirical procedure for creating data in Appendix B.1, and training the models in Appendix B.2. We also provided jupyter notebooks and custom python libraries that were used to create the plots and tables in the supplementary materials.

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

# A   THEORETICAL MODELS OF VARIABLE BINDING

In the appendix, we use the Dirac and Einstein summation conventions for brevity to derive all the theoretical expositions in the main paper.

## A.1   MATHEMATICAL PRELIMINARIES

The core concept of the episodic memory theory is basis change, the appropriate setting of the stored memories. Current notations lack the ability to adequately capture the nuances of basis change. Hence, we introduce abstract algebra notations typically used in theoretical physics literature to formally explain the variable binding mechanisms. We treat a *vector* as an abstract mathematical object invariant to basis changes. Vectors have *vector components* that are associated with the respective basis under consideration. We use Dirac notations to represent vector $v$ as - $|v\rangle = \sum_i v^i |e_i\rangle$. Here, the linearly independent collection of vectors $|e_i\rangle$ is the *basis* with respect to which the vector $|v\rangle$ has component $v^i \in \mathbb{R}$. Linear algebra states that a collection of basis vectors $|e_i\rangle$ has an associated collection of *basis covectors* $\langle e^i|$ defined such that $\langle e^i|e_j\rangle = \delta_{ij}$, where $\delta_{ij}$ is the Kronecker delta. This allows us to reformulate the vector components in terms of the vector itself as $|v\rangle = \sum_i \langle e^i|v\rangle |e_i\rangle$. We use the Einstein summation convention to omit the summation symbols wherever the summation is clear. Therefore, vector $|v\rangle$ written in basis $|e_i\rangle$ is

$$
\begin{aligned}
|v\rangle &= v^i |e_i\rangle \\
&= \langle e^i|v\rangle |e_i\rangle .
\end{aligned}
\tag{6}
$$

The set of all possible vectors $|v\rangle$ is a *vector space* spanned by the basis vectors $|e_i\rangle$. A *subspace* of this space is a vector space that is spanned by a subset of basis vectors $\{|e'_j\rangle : |e'_j\rangle \subseteq \{|e_i\rangle\}\}$.

The RNN dynamics presented in Equation 1 represented in the new notation is reformulated as:

$$
\begin{aligned}
|h(t)\rangle &= \sigma_f \left( \left( \xi^i_\mu \Phi^\mu_\nu (\xi^\dagger)^\nu_j \ |e_i\rangle \langle e^j| \right) |h(t-1)\rangle \right) \\
&= \sigma_f \left( W_{hh} \, \vec{h}(t-1) \right) .
\end{aligned}
\tag{7}
$$

The greek indices iterate over memory space dimensions $\{1, 2, \ldots, N_h\}$, alpha numeric indices iterate over feature dimension indices $\{1, 2, \ldots, N_f\}$. Typically, we use the standard basis in our simulations. For the rest of the paper, the standard basis will be represented by the collection of vectors $|e_i\rangle$ and the covectors $\langle e^i|$. The hidden state at time $t$ in the standard basis is denoted as $|h(t)\rangle = \langle e^j|h(t)\rangle |e_i\rangle$. $\langle e^j|h(t)\rangle$ are the *vector components* of $|h(t)\rangle$ we obtain from simulations.

## A.2   RNN-EPISODIC MEMORY EQUIVALENCE

We modify the GSEMM formulation with a pseudoinverse learning rule instead of the Hebbian learning rule for the synapses. This modification allows us to deal with more general (linearly independent vectors) memories than orthogonal vectors Personnaz et al. (1986). The dynamical equations for our modified GSEMM are given below.

$$
\begin{cases}
\mathcal{T}_f \dfrac{dV_f}{dt} = & \sqrt{\alpha_s} \, \Xi \, \sigma_h(V_h) - V_f, \\[2mm]
\mathcal{T}_h \dfrac{dV_h}{dt} = & \sqrt{\alpha_s} \, \Xi^\dagger \, \sigma_f(V_f) + \alpha_c \Phi'^\top \Xi^\dagger V_d - V_h, \\[2mm]
\mathcal{T}_d \dfrac{dV_d}{dt} = & \sigma_f(V_f) - V_d,
\end{cases}
\tag{8}
$$

The neural state variables of the dynamical system are $V_f \in \mathbb{R}^{N_f \times 1}, V_h \in \mathbb{R}^{N_h \times 1}, V_d \in \mathbb{R}^{N_f \times 1}$. The interactions are represented by $\Xi \in \mathbb{R}^{N_f \times N_h}$ and $\Phi \in \mathbb{R}^{N_h \times N_h}$. $\Xi^\dagger$ is the Moore-Penrose pseudoinverse of $\Xi$. To derive the connection between the continuous time model and discrete updates of RNNs, we discretize the continuous time model using forward Euler approximation under the conditions that $\mathcal{T}_f = 1, \mathcal{T}_h = 0$, and $\mathcal{T}_d = 0$. From a given time $t$, the update equations are given as

$$\begin{cases} \mathcal{T}_f(V_f(t+1) - V_f(t)) = & \Xi\,\sigma_h(V_h(t)) - V_f(t)\,, \\ V_h(t) = & \Xi^\top\,\sigma_f(V_f(t)) + \Phi^\top\Xi^\top V_d(t)\,, \\ V_d(t) = & \sigma_f(V_f(t))\,. \end{cases} \tag{9}$$

$$\begin{cases} \mathcal{T}_f(V_f(t+1) - V_f(t)) = & \Xi\,\sigma_h(V_h) - V_f(t)\,, \\ V_h(t) = & \Xi^\top\,\sigma_f(V_f(t)) + \Phi^\top\Xi^\top\sigma_f(V_f)\,, \end{cases} \tag{10}$$

$$\begin{cases} \mathcal{T}_f(V_f(t+1) - V_f(t)) = & \Xi\,V_h - V_f(t)\,, \\ V_h(t) = & (I + \Phi^\top)\Xi^\top\sigma_f(V_f)\,, \end{cases} \tag{11}$$

$$\mathcal{T}_f(V_f(t+1) - V_f(t)) = \Xi\,(I + \Phi^\top)\Xi^\top\sigma_f(V_f) - V_f(t) \tag{12}$$

Final discrete upate equation

$$V_f(t+1) = \Xi\,(I + \Phi^\top)\Xi^\top\sigma_f(V_f) \tag{13}$$

Restrict the norm of matrix $\|\Xi\,(I + \Phi^\top)\Xi^\top\| \le 1$.
This allows us to consider the transformation $V_f' = \sigma_f(V_f)$, so for invertible $\sigma_f$,

$$\sigma_f^{-1}(V_f'(t+1)) = \Xi\,(I + \Phi^\top)\Xi^\top V_f' \tag{14}$$

$$\sigma_f^{-1}(V_f'(t+1)) = \Xi\,(I + \Phi^\top)\Xi^\top V_f' \tag{15}$$

$$V_f'(t+1) = \sigma_f(\Xi\,(I + \Phi^\top)\Xi^\top V_f') \tag{16}$$

this is a general update equation for an RNN without bias. The physical interpretation of this equation is that the columns of $\Xi$ stores the individual *memories* of the system and the linear operator $(I + \Phi)$ is the temporal interaction between the stored *memories*. In the memory modeling literature, it is typical to consider memories as a fixed collection instead of a variable collection that shares a common interaction behavior. We will show how in the next sections how the dynamics as a result of fixed collection can be used to store variable information.

**Topological Conjugacy with RNNs**: Proof that dynamical systems governed by Equations 13 and 16 are topological conjugates.

Consider $f(x) = \Xi\,(I + \Phi^\top)\Xi^\top\sigma_f(x)$ for Equation 13 and $g(x) = \sigma_f(\Xi\,(I + \Phi^\top)\Xi^\top x)$ for Equation 16. Consider a homeomorphism $h(y) = \sigma_f(y)$ on $g$. Then,

$$\begin{aligned} (h^{-1} \circ g \circ h)(x) &= \sigma_f^{-1}(\sigma_f(\Xi\,(I + \Phi^\top)\Xi^\top\sigma_f(x))) \\ &= \Xi\,(I + \Phi^\top)\Xi^\top\sigma_f(x) \\ &= f(x) \end{aligned} \tag{17}$$

So, for the homeomorphism $h$ on $g$, we get that $h^{-1} \circ g \circ h = f$ proving that $f$ and $g$ are topological conjugates. Therefore all dynamical properties of $f$ and $g$ are shared.

### A.3 EXAMPLE: GENERALIZED FIBONACCI SERIES.

We consider a generalization of the Fibonacci sequence where each element $F_n \in \mathbb{R}^d$ is a vector defined recursively as,

$$F_n = \begin{cases} u^1 & n = 1 \\ u^2 & n = 2 \\ \vdots \\ u^s & n = s \\ \sum_{t=n-s}^{n-1} F_t & n > s \end{cases} \tag{18}$$

For any $u^i \in \mathbb{R}^d$. In order to store this *process* of generating sequences of $F_n$, the vectors $u_1, u_2, \ldots u_s$ needs to be stored as variables and recursively added to produce new outputs. In our framework, this can be accomplished by initializing the hidden state such that $|h(s)\rangle = \sum_i \sum_{\psi_\mu \in \{\Psi_i\}} u_i^\mu |\psi_\mu\rangle$, that is each $u^\mu$ is stored as activity of distinct subspaces of the hidden state. To encode the Fibonacci process in $\Phi$, we propose the following form for the inter-memory interactions.

$$\Phi = \sum_{\mu=1}^{(s-1)d} |\psi_\mu\rangle \langle \psi^{\mu+d}| + \underbrace{\left( \sum_{\mu=1}^{d} |\psi_{(s-1)d+\mu}\rangle \left( \sum_{\nu=0}^{s-1} \langle \psi^{\nu d+\mu}| \right) \right)}_{\sum_{t=n-s}^{n-1} F_t}. \tag{19}$$

This form of $\Phi$ has two parts. The first part implements a variable shift operation. The second part implements the summation function of Fibonacci. Since the hidden state is initialized with all the starting variables $u^\mu$, application of the $\Phi$ operator repeatedly, produces the next element in the sequence. As of now, the hidden state contains all the elements in the sequence. The abstract algebra notation allows proposing $W_r$ which will extract only the required output. Formally,

$$W_r = \Psi_s^* = \sum_{\mu=(s-1)d+1}^{sd} |e_{\mu-(s-1)d}\rangle \langle \psi^\mu|. \tag{20}$$

It is the projection operator which extracts the contents of the $N^{\text{th}}$ variable memory in the standard basis. Note that the process works irrespective of the actual values of $u^\mu$. To summarize, we now have a memory model encoding a generalizable process of fibonacci sequence generation.

## A.4 EXAMPLE: REPEAT COPY ($\mathcal{T}_1$)

Repeat Copy is a task typically used to evaluate the memory storage characteristics of RNNs since the task has a deterministic evolution represented by a simple algorithm that stores all input vectors in memory for later retrieval. Although elementary, repeat copy provides a simple framework to work out the variable binding circuit we theorized in action. For the repeat copy task, the linear operators of the RNN has the following equations.

$$\begin{cases} \Phi = \sum_{\mu=1}^{(s-1)d} |\psi_\mu\rangle \langle \psi^{\mu+d}| + \sum_{\mu=(s-1)d+1}^{sd} |\psi_\mu\rangle \langle \psi^{\mu-(s-1)d}| \\ W_{uh} = \Psi_s \\ W_r = \Psi_s^* \end{cases} \tag{21}$$

This $\phi$ can be imagined as copying the contents of the subspaces in a cyclic fashion. That is, the content of the $i^{th}$ subspace goes to $(i-1)^{\text{th}}$ subspace with the first subspace being copied to the $N^{\text{th}}$ subspace. The dynamical evolution of the RNN is represented at the time step 1 as,

$$|h(1)\rangle = |\psi_{(s-1)d+j}\rangle \langle e^j| u^i(1) |e_i\rangle \tag{22}$$

$$|h(1)\rangle = u^i(1) |\psi_{(s-1)d+j}\rangle \langle e^j|e_i\rangle \tag{23}$$

$$|h(1)\rangle = u^i(1) |\psi_{(s-1)d+j}\rangle \delta_{ij} \tag{24}$$

Kronecker delta index cancellation

$$|h(1)\rangle = u^i(1) |\psi_{(s-1)d+i}\rangle \tag{25}$$

At time step 2,

$$|h(2)\rangle = u^i(1) \Phi |\psi_{(s-1)d+i}\rangle + u^i(2) |\psi_{(s-1)d+i}\rangle \tag{26}$$

Expanding $\Phi$

$$|h(2)\rangle = u^i(1) \left( \sum_{\mu=1}^{(s-1)d} |\psi_\mu\rangle \langle\psi^{\mu+d}| + \sum_{\mu=(s-1)d+1}^{sd} |\psi_\mu\rangle \langle\psi^{\mu-(s-1)d}| \right) |\psi_{(s-1)d+i}\rangle \quad (27)$$
$$+ u^i(2) |\psi_{(s-1)d+i}\rangle$$

$$|h(2)\rangle = u^i(1) |\psi_{(s-2)d+i}\rangle + u^i(2) |\psi_{(s-1)d+i}\rangle \quad (28)$$

At the final step of the input phase when $t = s$, $|h(s)\rangle$ is defined as:

$$|h(s)\rangle = \sum_{\mu=1}^{s} u^i(\mu) |\psi_{(\mu-1)d+i}\rangle \quad (29)$$

For $t$ timesteps after $s$, the general equation for $|h(s+t)\rangle$ is:

$$|h(s+t)\rangle = \sum_{\mu=1}^{s} u^i(\mu) |\psi_{[((\mu-t-1 \mod s)+1)d+i]}\rangle \quad (30)$$

From this equation for the hidden state vector, it can be easily seen that the $\mu^{\text{th}}$ variable is stored in the $[(\mu - t - 1 \mod s) + 1]^{\text{th}}$ subspace at time step $t$. The readout weights $W_r = \Psi_s^*$ reads out the contents of the $s^{\text{th}}$ subspace.

## A.5 APPLICATION TO GENERAL RNNS

The linear RNNs we discussed are powerful in terms of the content of variables that can be stored and reliably retrieved. The variable contents, $u^i$, can be any real number and this information can be reliably retrieved in the end using the appropriate readout weights. However, learning such a system is difficult using gradient descent procedures. To see this, setting the components of $\Phi$ to anything other than unity might result in dynamics that is eventually converging or diverging resulting in a loss of information in these variables. Additionally, linear systems are not used in the practical design of RNNs. The main difference is now the presence of the nonlinearity. In this case, our theory can still be used. To illustrate this, consider a general RNN evolving according to $h(t+1) = g(W_{hh}h(t)+b)$ where $b$ is a bias term. Suppose $h(t) = h^*$ is a fixed point of the system. We can then linearize the system around the fixed point to obtain the linearized dynamics in a small region around the fixed point.

$$h(t + 1) - h^* = \mathcal{J}(g)|_{h^*} W_{hh} (h(t+1) - h^*) + O((h(t+1) - h^*)^2) \quad (31)$$

where $\mathcal{J}$ is the jacobian of the activation function $g$. If the RNN had an additional input, this can also be incorporated into the linearized system by treating the external input as a control variable

$$h(t + 1) - h^* = \mathcal{J}(g)|_{h^*} W_{hh} (h(t) - h^*) + \mathcal{J}(g)|_{h^*} W_{uh} u(t) \quad (32)$$

Substituting $h(t) - h^* = h'(t)$

$$h'(t + 1) = \mathcal{J}(g)|_{h^*} W_{hh} h'(t) + \mathcal{J}(g)|_{h^*} W_{uh} u(t) \quad (33)$$

which is exactly the linear system which we studied where instead of $W_{hh} = \Xi\Phi\Xi^\dagger$, we have $J(g)|_{h^*} W_{hh} = \Xi\Phi\Xi^\dagger$.

## A.6 ERROR ANALYSIS OF THE VARIABLE MEMORY APPROXIMATION ALGORITHM

Our empirical results revealed that there are certain cases of tasks where the algorithm fails to retrieve the correct basis transformation. In this section, we will investigate why this dissociation from theory happens. To this end, we want to formalize and compare what the hidden state is according to the power iteration ($h(t)$) and the variable memories ($|h(t)\rangle$).

$$|h(0)\rangle = 0 \qquad h(0) = 0 \tag{34}$$

$$|h(1)\rangle = u^i(1) |\psi_i\rangle \qquad h(1) = W_r^\dagger u(1) \tag{35}$$

$$|h(1)\rangle = u^i(1) \left|\psi_{(d+i)}\right\rangle + (u^i(2) + \Phi(0,\dots,u(1))^i) |\psi_i\rangle \qquad h(1) = W_{hh}W_r^\dagger u(1) + W_r^\dagger u(2) \tag{36}$$

The error in the basis definition is given by

$$\tilde{\Psi}_2 - \Psi_2 = \Phi(0,\dots u(1))^i |\psi_i\rangle \langle e^j| \tag{37}$$

$$\tilde{\Psi}_3 - \Psi_3 = \Phi(0,\dots u(1))^i \left|\psi_{(d+i)}\right\rangle \langle e^j| + \Phi(0,\dots u(2), u(1))^i \left|\psi_{(i)}\right\rangle \langle e^j| \tag{38}$$

For the power iteration to succeed in giving the correct variable memories, the effect of $\Phi$ acting on the variable memories needs to be negligible. In the case of repeat copy, this effect is zero as the operator $\Phi$ does not utilize any of the variables until the end of the input phase. For some of the compose copy tasks, we showed in the paper, this effect is negligible. Another way to think about this issue is that the $\Phi$ keeps on acting on the variable memories during the input phase and produces outputs even though the variables are not filled in fully yet. This behavior *pollutes* the definition of variable memories using power iteration. If $\Phi$ is sufficiently representative, for instance, the operator associated with the Fibonacci generation, then after the first input is passed, the 2nd variable memory will be the sum of the 1st and 2nd variable memories. Future development to the algorithm to general tasks needs to take this figure out ways to get over this error.

## B EXPERIMENTS

### B.1 DATA

We train RNNs on the variable binding tasks described in the main paper with the following restrictions - the domain of $u$ at each timestep is binary $\in \{-1, 1\}$ and the function $f$ is a linear function of its inputs. We collect various trajectories of the system evolving accoding to $f$ by first sampling uniformly randomly, the input vectors. The system is then allowed to evolve with the recurrent function $f$ over the time horizon defined by the training algorithm.

### B.2 TRAINING DETAILS

**Architecture**  We used single layer Elman-style RNNs for all the variable binding tasks. Given an input sequence $(u(1), u(2), ..., u(T))$ with each $u(t) \in \mathbb{R}^d$, an Elman RNN produces the output sequence $y = (y(1), ..., y(T))$ with $y(t) \in \mathbb{R}^{N_{out}}$ following the equations

$$h(t+1) = \tanh(W_{hh}h(t) + W_{uh}u(t)) \quad , \quad y(t) = W_r h(t) \tag{39}$$

Here $W_{uh} \in \mathbb{R}^{N_h \times N_{in}}$, $W_{hh} \in \mathbb{R}^{N_h \times N_h}$, and $W_r \in \mathbb{R}^{N_{out} \times N_h}$ are the input, hidden, and readout weights, respectively, and $N_h$ is the dimension of the hidden state space.

The initial hidden state $h(0)$ for each model was *not* a trained parameter; instead, these vectors were simply generated for each model and fixed throughout training. We used the zero vector for all the models.

**Task Dimensions**  Our results presented in the main paper for the repeat copy ($\mathcal{T}_1$) and compose copy ($\mathcal{T}_2$) used vectors of dimension $d = 8$ and sequences of $s = 8$ vectors to be input to the model.

**Training Parameters**  We used PyTorch's implementation of these RNN models and trained them using Adam optimization with MSE loss. We performed a hyperparameter search for the best parameters for each task — see table 2 for a list of our parameters for each task.

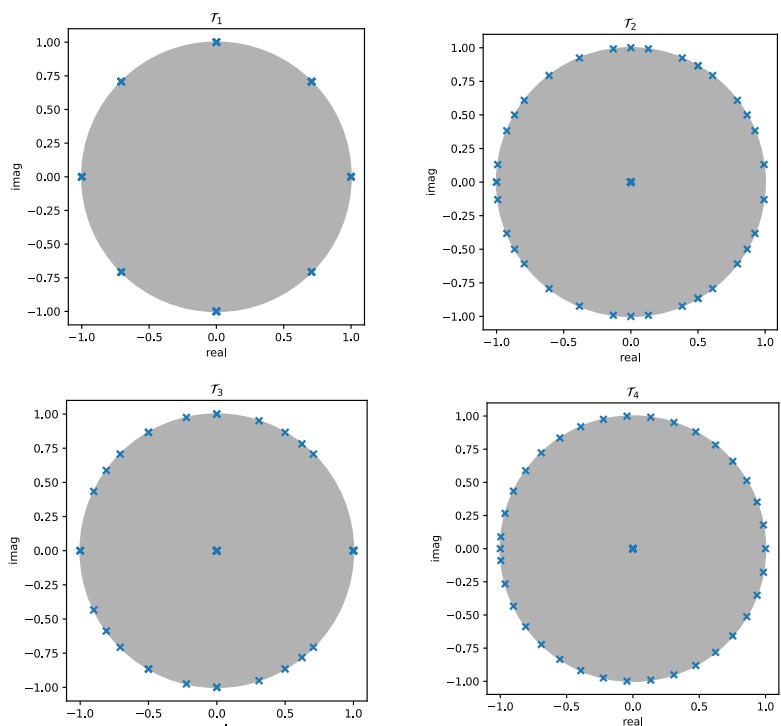

Figure 5: **Diverse range of high-dimensional dynamical behavior around the fixed point for the variable binding tasks**: The figures show the distribution of the eigenspectrum of the Jacobian around origin for the Elman RNN in the complex plane. For each of the representative tasks, the spectral analysis reveals a very high dimensional dynamical behavior with a complex spectral distribution. The dynamical behavior is non-trivial to interpret by the analysis of the Jacobian spectrum in the class of variable binding tasks alone. The EMT provides an alternate interpretation of the spectrum in these cases.

| | Repeat Copy ($\mathcal{T}_1$) | Compose Copy ($\mathcal{T}_2$) |
|---|---|---|
| Input & output dimensions | 8 | 8 |
| Input phase (# of timesteps) | 8 | 8 |
| training horizon | 100 | 100 |
| Hidden dimension $N_h$ | 128 | 128 |
| # of training iterations | 45000 | 45000 |
| (Mini)batch size | 64 | 64 |
| Learning rate | $10^{-3}$ | $10^{-3}$ |
| applied at iteration | 36000 | |
| Weight decay ($L^2$ regularization) | none | none |
| Gradient clipping threshold | 1.0 | 1.0 |

Table 2: Architecture, Task, & Training Parameters

**Curriculum Time Horizon**     When training a network, we adaptively adjusted the number of timesteps after the input phase during which the RNN's output was evaluated. We refer to this window as the *training horizon* for the model.

Specifically, during training we kept a rolling average of the model's *loss by timestep* $L(t)$, i.e. the accuracy of the model's predictions on the $t$-th timestep after the input phase. This metric was computed from the network's loss on each batch, so tracking $L(t)$ required minimal extra computation.

The network was initially trained at time horizon $H_0$ and we adapted this horizon on each training iteration based on the model's loss by timestep. Letting $H_n$ denote the time horizon used on training step $n$, the horizon was increased by a factor of $\gamma = 1.2$ (e.g. $H_{n+1} \leftarrow \gamma H_n$) whenever the model's accuracy $L(t)$ for $t \leq H_{\min}$ decreased below a threshold $\epsilon = 3 \cdot 10^{-2}$. Similarly, the horizon was

reduced by a factor of $\gamma$ is the model's loss was above the threshold ($H_{n+1} \leftarrow H_n/\gamma$). We also restricted $H_n$ to be within a minimum training horizon $H_0$ and maximum horizon $H_{\max}$. These where set to 10/100 for the repeat copy task and 10/100 for the compose copy task.

We found this algorithm did not affect the results presented in this paper, but it did improve training efficiency, allowing us to run the experiments for more seeds.

### B.3 REPEAT COPY: FURTHER EXAMPLES OF HIDDEN WEIGHTS DECOMPOSITION

This section includes additional examples of the hidden weights decomposition applied to networks trained on the repeat copy task.

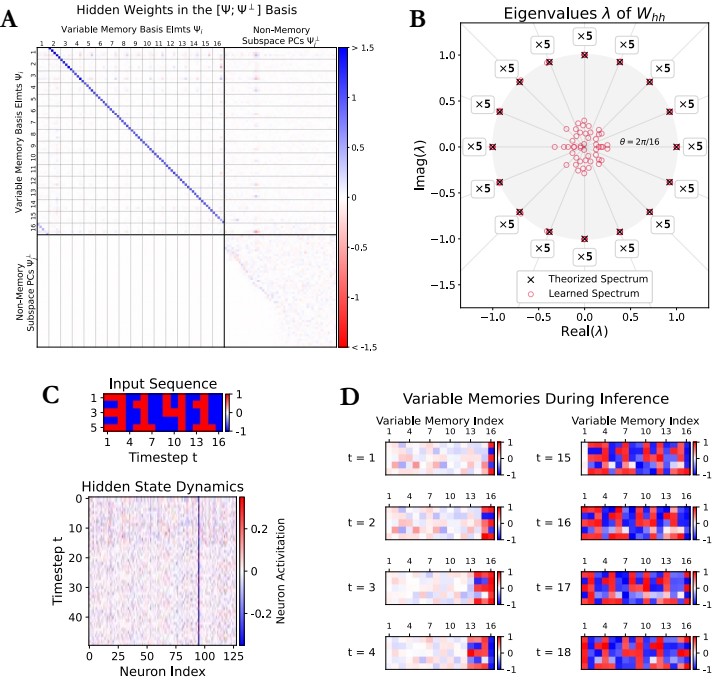

Figure 6: **Additional Experimental Results of Repeat Copy Task with 16 vectors, each of 5 dimensions**: **A**. $W_{hh}$ visualized in the variable memory basis reveals the variable memories and their interactions. **B**. After training, the eigenspectrum of $W_{hh}$ with a magnitude $\geq 1$ overlaps with the theoretical $\Phi$. The boxes show the number of eigenvalues in each direction. **C**. During inference, "3141" is inserted into the network in the form of binary vectors. This input results in the hidden state evolving in the standard basis, as shown. How this hidden state evolution is related to the computations cannot be interpreted easily in this basis. **D**. When projected on the variable memories, the hidden state evolution reveals the contents of the variables over time. Note that in order to make these visualization clear, we needed to normalize the activity along each variable memory to have standard deviation 1 when assigning colors to the pixels. The standard deviation of the memory subspaces varies due to variance in the strength of some variable memory interactions. These differences in interaction strengths does not impede the model's performance, however, likely due to the nonlinearity of the activation function. Unlike the linear model, interaction strengths well above 1 cannot cause hidden state space to expand indefinitely because the tanh nonlinearity restricts the network's state to $[-1, 1]^{N_h}$. This property appears to allow the RNN to sustain stable periodic cycles for a range of interaction strengths above 1.

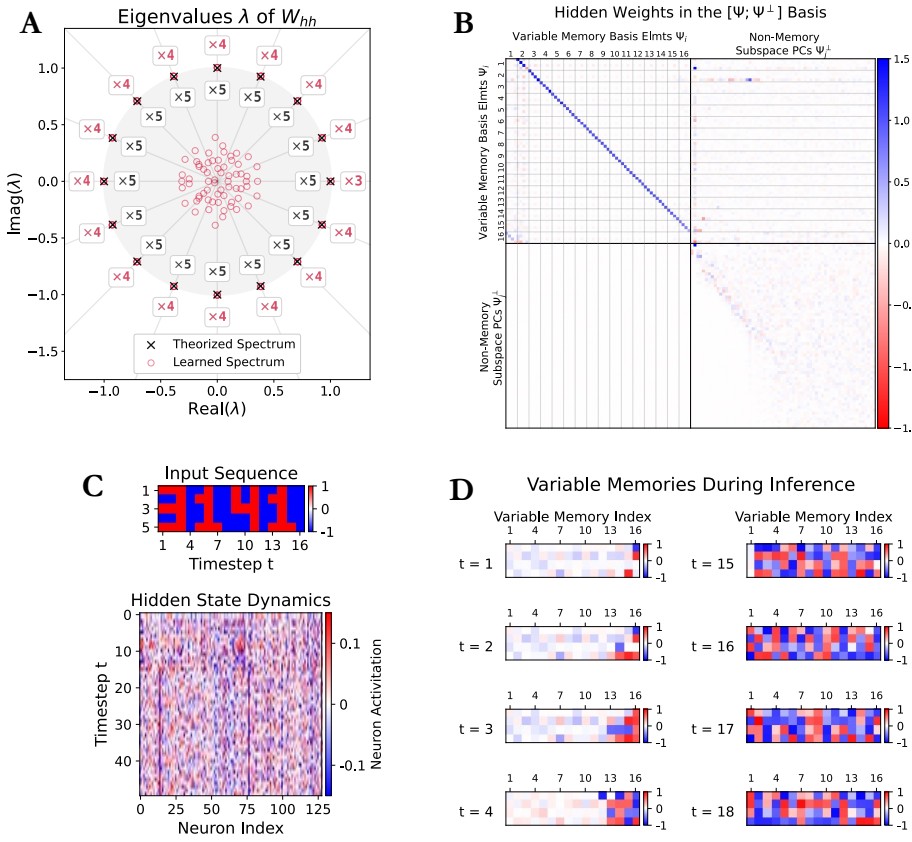

Figure 7: **Nonlinear Variable Memories Learned for the Repeat Copy Task with 16 vectors, each of 5 dimensions**. **A**. Eigenspectrum of $W_{hh}$ after training. The learned eigenvalues cluster into 16 groups equally spaced around the unit circle, but there are only 3-4 eigenvectors per cluster (indicated in red). Compare this to the theorized linear model, which has 5 eigenvalues per cluster (indicated in black). The task requires $16 \cdot = 5 = 80$ bits of information to be stored. Linearization about the origin predicts that the long-term behavior of the model is dictated by the eigenvectors with eigenvalue outside the unit circle because its activity along other dimensions will decay over time. The model has only $16 \cdot 4 - 1 = 63$ eigenvectors with eigenvalue near the unit circle, so this results suggests the model has learned a non-linear encoding that compresses 80 bits of information into 63 dimensions. **B**: $W_{hh}$ visualized in the variable memory basis reveals the variable memories and their interactions. Here, the variable memories have only 4 dimensions because the network has learned only 63 eigenvectors with eigenvalue near the unit circle. The variable memory subspaces also have non-trivial interaction with a few of the the non-memory subspaces. **C**. During inference, "3141" is inserted into the network in the form of binary vectors. This input results in the hidden state evolving in the standard basis, as shown. How this hidden state evolution is related to the computations cannot be interpreted easily in this basis. **D**. When projected on the variable memories, the hidden state evolution is still not easily interpreted for this network, likely due to a nonlinear variable memories. As in the previous figure, we normalized the activity along each variable memory to have standard deviation 1 when assigning colors to the pixels.

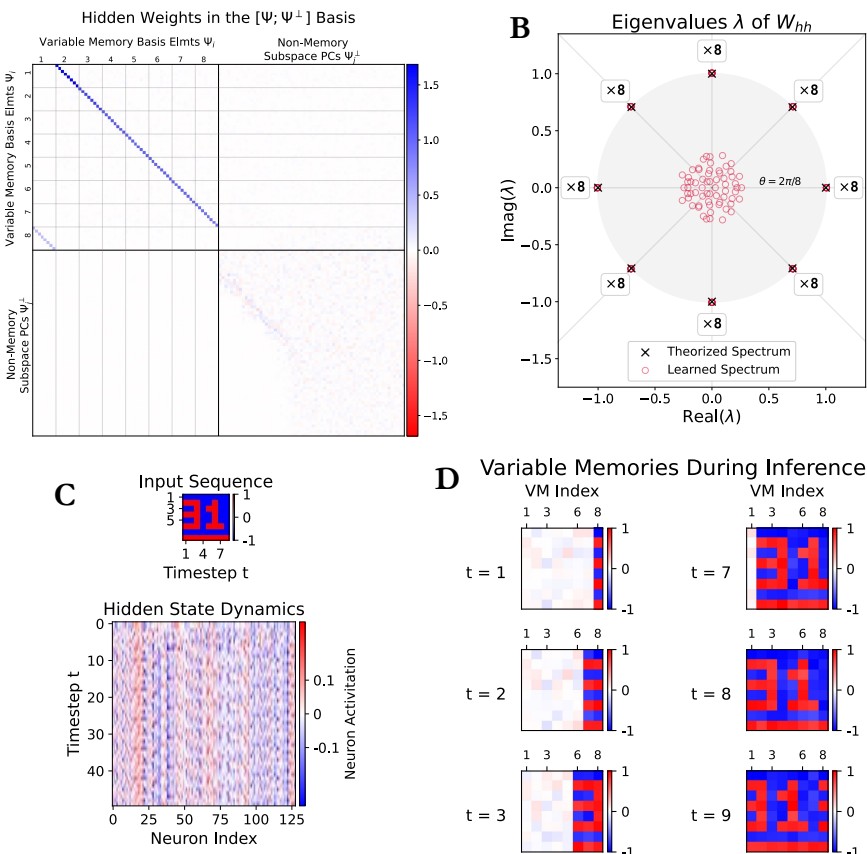

Figure 8: **Additional Experimental Results of Repeat Copy Task with 8 vectors, each of 8 dimensions**. This figure was included to show the decomposition applied to other values of $s$ and $d$ for the Repeat Copy task. **A**: $W_{hh}$ visualized in the variable memory basis reveals the variable memories and their interactions. **B**. After training, the eigenspectrum of $W_{hh}$ with a magnitude $\geq 1$ overlaps with the theoretical $\Phi$. The boxes show the number of eigenvalues in each direction. **C**. During inference, "3141" is inserted into the network in the form of binary vectors. This input results in the hidden state evolving in the standard basis, as shown. How this hidden state evolution is related to the computations cannot be interpreted easily in this basis. **D**. When projected on the variable memories, the hidden state evolution reveals the contents of the variables over time. As in the previous figures, we normalized the activity along each variable memory to have standard deviation 1 when assigning colors to the pixels.

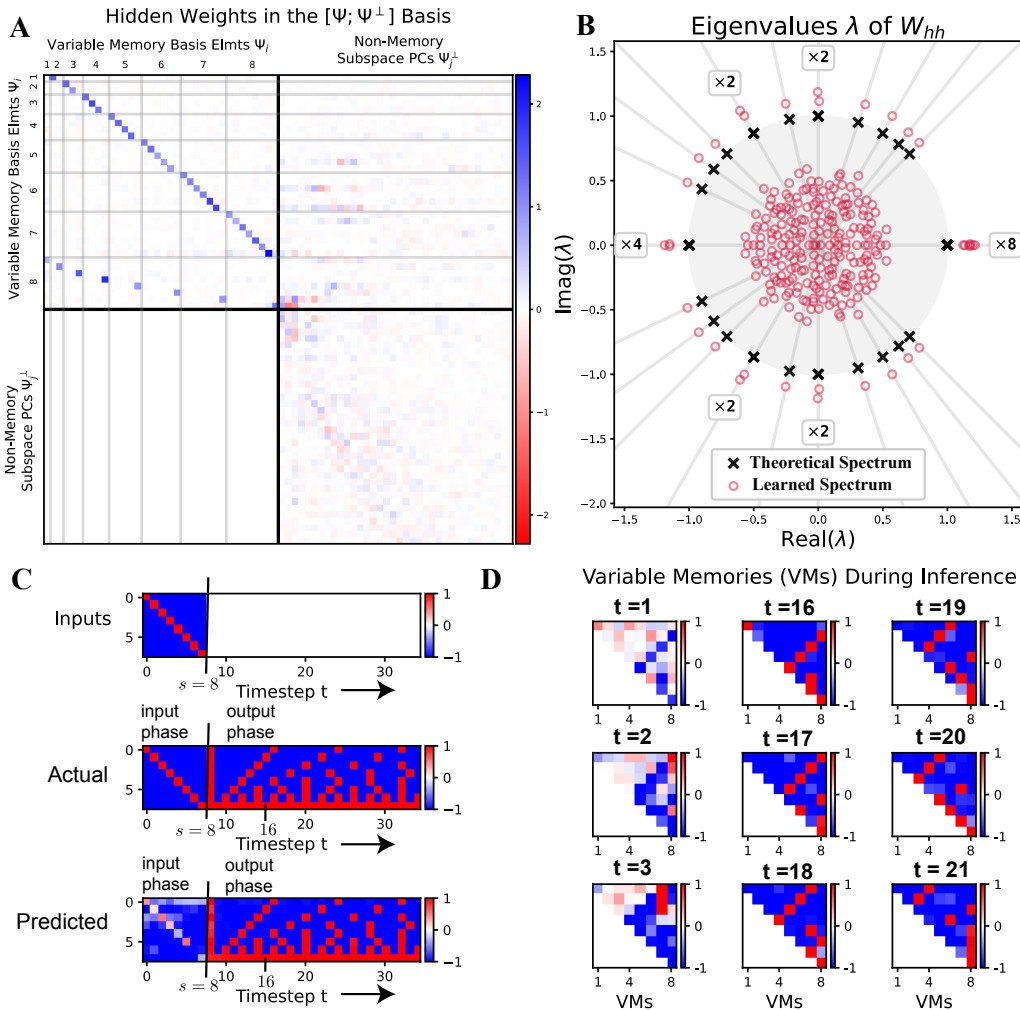

Figure 9: **Experimental Results of Compose Copy Task ($\mathcal{T}_3$) of 8 vectors, each of 8 dimensions**:
**A**. $W_{hh}$ visualized in the variable memory basis reveals the variable memories and their interactions. It is observed that $W_{hh}$ encodes an optimized version of the theoretical mechanisms since there are dimensions in the variable memories irrelevant for future computations. **B**. Compared to repeat copy, the eigenspectrum of the learned $W_{hh}$ is more complex, yet the theoretical $\Phi$ accurately predicts the angles and number of eigenvalues. The eigenvalues' magnitude greater than 1 (rather than close to 1 found in repeat copy) indicate that the non-linearity plays a role in controlling the diverging behavior of the spaces. **C**. During the output phase, the past $s$ variables are composed to form future outputs. **D**. The hidden state evolution, when projected on the variable memories, reveals the contents of the variables over time. Unlike repeat copy, the input phase does not precisely match the model's theoretical predictions. Transient dynamics dominate the initial timesteps, clouding the underlying computations (t=1, 2, 3). Yet the long-term behavior (from t=16) of the output phase behavior is as the theoretical model predicts with the composed result stored in the $8^{\text{th}}$ variable memory at each time.

### B.4 Uniform vs. Gaussian Parameter Initialization

We also tested a different initialization scheme for the parameters $W_{uh}, W_{hh}$, and $W_{hy}$ of the RNNs to observe the effect(s) this would have on the structure of the learned weights. The results presented in the main paper and in earlier sections of the Supplemental Material used PyTorch's default initialization scheme: each weight is drawn *uniformly* from $[-k, k]$ with $k = 1/\sqrt{N_h}$. Fig. 10 shows the resulting spectrum of a trained model when it's parameters where drawn from a Gaussian distribution with mean 0 and variance $1/N_h$. One can see that this model learned a spectrum similar to that presented in the main paper, but the largest eigenvalues are further away from the unit circle. This result was observed for most seeds for networks trained on the repeat copy task with $s = 8$ vectors of dimension $d = 4$ and $d = 8$, though it doesn't hold for every seed. We also find that the networks whose spectrum has larger eigenvalues usually generalize longer in time than the networks with eigenvalues closer to the unit circle.

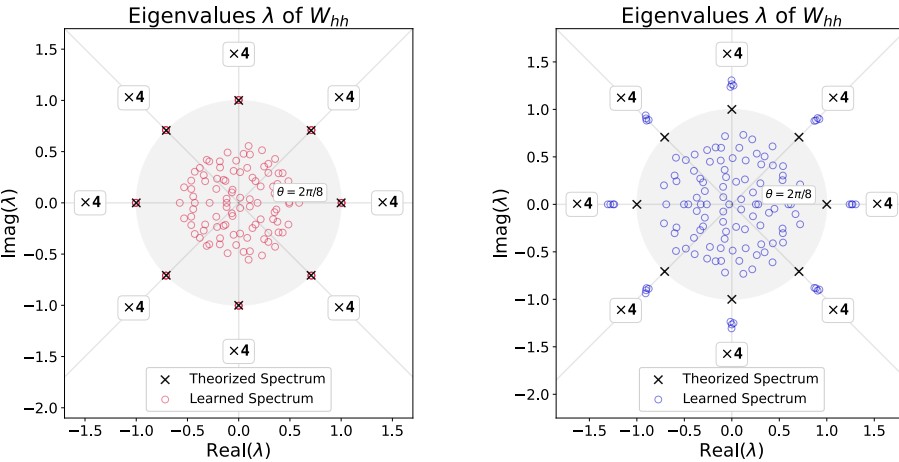

(a) Uniform Initialization          (b) Gaussian Initialization

Figure 10: **An effect of parameter initialization for the Repeat Copy Task** with $s = 8$ vectors, each of dimension $d = 4$. **A**: Spectrum (in red) of the learned hidden weights $W_{hh}$ for a network whose parameters where initialized from a uniform distribution over $[-k, k]$ with $k = 1/\sqrt{N_h}$. This network has 32 eigenvalues that are nearly on the unit circle. These eigenvalues are clustered into groups of 4, each group being an angle of $\theta = 2\pi/s$ apart from each other. These eigenvalues coincide with the eigenvalues of the linear model for solving the repeat copy task. **B**: Spectrum (in blue) of the learned hidden weights $W_{hh}$ for a network whose parameters where initialized from a Gaussian distribution with mean 0 and variance $1/N_h$. This network has 32 eigenvalues outside the unit circle, but they are a larger radii than the model initialized using the uniform distribution. These eigenvalues still cluster into eight groups of four, and the average complex argument of each group aligns with the complex arguments of the eigenvalues for the linear model.

