# OpenReview forum: "Episodic Memory Theory for the Mechanistic Interpretation of Recurrent Neural Networks"
_ICLR.cc/2024/Conference — Submitted to ICLR 2024_

### Official Review · Reviewer_VSBc · 2023-10-26

**Soundness:** 2 fair
**Presentation:** 1 poor
**Contribution:** 1 poor
**Rating:** 1
**Confidence:** 4

**Summary:**

The authors employed a linear RNN to execute a task called VARIABLE BINDING. They utilized linear algebra to illustrate the process of extracting outputs from the network.

**Strengths:**

The specific sections of the article that describe what was done are relatively clear.

**Weaknesses:**

There are three main drawbacks:

1. The current analysis is overly limited. It applies solely to a single-layer RNN with linear dynamics. The assertion that any general RNN can be treated as a linear RNN in A.4 is fundamentally incorrect. In reality, a general RNN may not behave near a fixed point, resulting in significant higher-order terms. Furthermore, the analysis pertains to a task, the VARIABLE BINDING TASK, which is notably distinct from a translation task.

2. The current analysis lacks novelty. The article primarily employs linear projection techniques to examine the components of RNN weights, a methodology that has been in use for a considerable period.

3. The presentation lacks clarity. Figures 1 and 2 are difficult to comprehend, and their intended message is unclear. Additionally, the mention of GSEMM seems unnecessary since the current work solely involves a simple linear RNN. There is a lack of a concise summary of the core mechanisms of the RNN.

**Questions:**

See weakness.

---

> ### Author Response · Authors · 2023-11-21
>
> We thank the reviewer for the comments.
>
> "The current analysis is overly limited. ..."
>
> Our analysis is restricted to linear RNNs, but linearization approaches allow researchers to study complex non-linear dynamical systems like RNNs. What we described in A.4 is only one approach to linearization typically used in the literature. Alternate approaches, like the Koopman theory, allow a more complicated linearization while maintaining applicability in regions farther from fixed points. Our theory extends to these spaces, although we find in the paper that the variable binding tasks are encoded in a simple inner product (the finite-dimensional real vector inner product) space.
>
> “, the analysis pertains to a task, the VARIABLE BINDING TASK, which is notably distinct from a translation task”
>
> The class of VARIABLE BINDING TASKS is analogous to a deterministic version of s-gram tasks used in NLP. We only restrict the composition function to be linear. Future work will explore more complex non-linear composition functions.
>
> 1.	The current analysis lacks novelty. The article primarily employs linear projection techniques to examine the components of RNN weights, a methodology that has been in use for a considerable period.
>
> In our restriction of the VARIABLE BINDING TASKS, we find that a linear basis transformation reveals the operator. To our knowledge, an explicit operator that enables memory storage mechanisms in RNNs has not yet been proposed and experimentally validated. We further showed that the operator is not just a theoretical construct but emerges naturally after training. We also want to note that Fig 4,5 has not been extracted from prior RNN analyses.
>
> Simple tasks enable researchers to probe the mechanisms of complex dynamical systems like RNNs. For instance, simple tasks like 3-bit flip flop, Frequency cued sine wave, Context-dependent integration are used to probe various dynamical behaviors of RNNs [3]. Our tasks are more complex than these as the dynamical behavior for VARIABLE BINDING is very high dimensional and inscrutable (Appendix Figure 5 shows how high dimensional and diverse the Jacobian spectrum is after linearizing around the origin).
>
> Among the VARIABLE BINDING tasks, repeat copy ($T_1$) has been used frequently to analyze the memory storage behavior of RNNs [1]. Recently, the operator we proposed for repeat copy was also found in connection to traveling waves in RNNs [2], suggesting the theory's applicability is broader than the VARIABLE BINDING tasks.
>
>
> [1] Alex Graves, Greg Wayne, and Ivo Danihelka. Neural turing machines. ArXiv, abs/1410.5401, 2014
> [2] Thomas Anderson Keller, Lyle E. Muller, Terrence J. Sejnowski, and Max Welling. Traveling waves
> encode the recent past and enhance sequence learning. ArXiv, abs/2309.08045, 2023. URL https://api.semanticscholar.org/CorpusID:262013982
> [3] Maheswaranathan, Niru et al. “Universality and individuality in neural dynamics across large populations of recurrent networks.” Advances in neural information processing systems 2019 (2019): 15629-15641 .
>
> “Additionally, mentioning GSEMM seems unnecessary since the current work solely involves a simple linear RNN. There is a lack of a concise summary of the core mechanisms of the RNN.”
>
> GSEMM is not unnecessary, as this connection to neurocomputational memory models enabled interpreting the weights of the RNN as stored memories and their interactions. To our knowledge, this is the first connection between Hopfield-like memory models studied in theoretical neuroscience and Recurrent Neural Networks.
>
> "There is a lack of a concise summary of the core mechanisms of the RNN."
>
> Section 5 and Figure 2 summarize the main parts of the theory. Please let us know if any particular aspects of the figures and exposition are unclear.

---

> > ### Comment · Reviewer_VSBc · 2023-11-22
> >
> > Thanks for your detailed answer, while I believe there is still much room for improvement.

---

### Official Review · Reviewer_PqoU · 2023-10-30

**Soundness:** 2 fair
**Presentation:** 3 good
**Contribution:** 3 good
**Rating:** 6
**Confidence:** 3

**Summary:**

This paper develops the Episodic Memory Theory (EMT), where a circuit mechanism is presented to illustrate how a linear RNN recursively stores and composes hidden variables. The authors show that, under specially designed algorithmic tasks called *variable binding*, the hidden neurons and the learned parameters of a trained linear RNN can be illustrated by *variable memories* $\Psi$, which are a group of interpretable bases. They also design an operator $\Phi$ to form a circuit computation of variable memories $\Psi$. Finally, they propose a power iteration-based algorithm to find the bases $\Psi$ via the learned RNN parameters. In the experiment, the authors show the variable memories $\Psi$ could reveal the information stored in the hidden states of a linear RNN. They also provide examples of how such bases $\Psi$ enable human interpretability of learned RNN parameters.

**Strengths:**

* Proposes a novel basis (variable memory $\Psi$) that, for the first time, reveals hidden neurons actively involved in information processing in a linear RNN.

* Using the basis to interpret the learned parameters of a linear RNN in a  human-friendly way.

**Weaknesses:**

* Only a Repeat Copy task is shown to reveal the stored information in hidden neurons. As the authors mentioned in section 7.3, there are some cases in which the computed basis is converged, but it cannot give interpretable representations. In other words, under which tasks do we expect this framework to fail?

* Typos: The first sentence below Equation 6 should be "Figure 2A".

**Questions:**

* > This deviation from the theory is a result of the sensitivity of the basis definition to minor errors in the pseudo-inverse required to compute the dual.

* What is the meaning of "dual"? is this related to the conjugate transpose computation $EE^{*}$ in Algorithm 1?

* Could you please discuss in detail the difficulties of applying the proposed theory to nonlinear RNNs? In Appendix A.4, you have shown that a nonlinear RNN has a similar form of the linear system as linear RNNs instead of a different $W_{hh}$.

---

> ### Author Response · Authors · 2023-11-21
>
> We thank the reviewer for the comments and for pointing out the typo.
>
> •	Only a Repeat Copy task is shown to reveal the stored information in hidden neurons. As the authors mentioned in section 7.3, there are some cases in which the computed basis is converged, but it cannot give interpretable representations. In other words, under which tasks do we expect this framework to fail?
>
> We have added a section in the Appendix A.6 for a more formal analysis of the error in approximating variable memories. In short, the algorithm assumes specific interactions of the variable memory are negligible. This assumption is valid for the cases of repeat copy and compose copy we showed in the paper, but for some $f$ where this assumption is broken, the algorithm fails. We want to note that the failure of the algorithm does not mean that the theoretical mechanisms are not present (the convergence of the theoretical $\Phi$ in Table 1 still stands). This, however, means that it is not trivial to formulate an algorithm to find such a human interpretable basis for a general class of tasks.
>
> •	This deviation from the theory is a result of the sensitivity of the basis definition to minor errors in the pseudo-inverse required to compute the dual.
>
> We were incorrect on this point in the paper. More recent analysis has revealed that the deviation results from the errors accumulated when approximating the variable memories by power iteration. A section has been added to Appendix A.6 that formally explains the approximation error of the variable memory algorithm.
>
> •	What is the meaning of "dual"? is this related to the conjugate transpose computation ��∗ in Algorithm 1?
>
> "dual" is a misnomer. We meant the projection operator that extracts the components in the variable memory space into the standard basis. This is updated in the manuscript.
>
> •	Could you please discuss in detail the difficulties of applying the proposed theory to nonlinear RNNs? In Appendix A.4, you have shown that a nonlinear RNN has a similar form of the linear system as linear RNNs instead of a different �ℎℎ.
>
> The proposed algorithm (not the theory) does not work for non-linear RNNs when any of the two assumptions fail – (1) The geometry of the representation is not easily captured by a linear basis, in which case a more complex set of basis vectors needs to be taken, (2) If the RNN dynamics are far from the fixed point, which means that the Taylor series expansion in Appendix A.4 needs to account for the higher order terms, even if the correction from the Jacobian is taken into account. (1) and (2) are not mutually exclusive; sometimes failure of (2) implies the failure of (1).

---

### Official Review · Reviewer_xCsT · 2023-10-30

**Soundness:** 2 fair
**Presentation:** 2 fair
**Contribution:** 2 fair
**Rating:** 5
**Confidence:** 3

**Summary:**

The authors frame RNNs as episodic memory retrievers and use this to devise a circuit mechanism that could carry out sequential memory tasks. They show that this circuit mechanism seems to appear in trained networks.

**Strengths:**

The paper attempts to bring several different topic areas together, which is admirable.

The potential applications listed for this work would be useful if achievable.

 The work is thorough.

**Weaknesses:**

I struggled to read this paper at several points. It is pulling concepts from many different fields and also I believe trying to introduce new ones. I'm also not well versed in the specific notation used. I don't want to down-score work for being too interdisciplinary, but as the paper stands now I don't know if there is a large community who would be able to understand and benefit from it as a whole.

The tasks (insofar as they are described) seem like weak, or at least very specific, tests of variable binding. For the authors to make claims about variable binding in general, they would need to show tasks that do more than just require sequential repeats of the input.



A substantial issue for me is that I am confused about the elements that the authors label as being novel here. Most of them are, at least at a broad level, well-represented in the neuroscience-inspired RNN literature. For example, the authors say in the discussion:

we provide "a novel perspective on Recurrent Neural Networks (RNNs), framing them as dy-
namical systems performing sequence memory retrieval". The original Ellman model itself uses RNNs for a form of sequence memory retrieval, but also several more recent works study serial working memory with RNNs such as: https://direct.mit.edu/neco/article/30/6/1449/8400/A-Theory-of-Sequence-Indexing-and-Working-Memory and https://psycnet.apa.org/record/2006-04733-001

"We introduced the concept of “variable
memories,” linear subspaces capable of symbolically binding and recursively composing informa-
tion."  The notion of storing different items in different linear subspaces has also been explored: https://www.science.org/doi/10.1126/science.abm0204

"We presented a new class of algorithmic tasks that are designed to probe the variable binding
behavior of RNNs. " As represented by the above studies on serial working memory, this class of tasks is not new.

"for the first time, revealed hidden neurons actively involved in in-
formation processing in RNNs. " This is obviously not the first time people have studied how neurons process information in RNNs (see the work of Omri Barak and David Sussillo, e.g.).

On the whole I also don't see the specific value in claiming that this analysis is related to episodic memory. Sequential memory, yes. But there is nothing specifically episodic about the motivation for the analyses and the tasks represent serial working memory.

**Questions:**

What are the tasks? One is described in the main text and another mentioned in the appendix. All 4 should be described in the main text.

I thought u(t) was the input vector, which is 0 for t>s, yet Eqn 2 shows the evolution of u(t) for t>s.

The authors say:

"The mechanistic interpretability seeks to reverse-engineer neural net-
works to expose the underlying mechanisms enabling them to learn and adapt to previously unen-
countered conditions"

and

" This assumption limits the mod-
els’ applicability to mechanistic interpretability, which requires the symbolic binding of memories
typically available only during inference."

Why are they focusing on mechanistic interpretability for such a limited behavior? As I understand it MI can be used to explain any behavior of a neural network.

---

> ### Author Response · Authors · 2023-11-21
>
> We thank the reviewer for the insightful comments and for pointing us to the relevant literature.
>
> "The tasks (insofar as they are described) seem like weak, or at least very specific, tests of variable binding. ..."
>
> Without the linear assumption, the variable binding tasks can represent any dynamic system with a historical context of s states or, in NLP terms, an s-gram model. Probablistic versions of s-gram models have been used in the literature before; we take a restricted $f$ for better analytic tractability.
>
> Variable binding is a very diverse phenomenon, and tackling all of its aspects in a single paper is intractable. Our class of tasks is a simple restriction that can be used to probe some mechanisms of variable binding. Although simple, these tasks reveal the mechanisms of RNNs and contribute to developing a better, more complete theory of variable binding.
>
> We also note that simple tasks are used liberally in the literature to improve our understanding of RNN computations. Starting with small-scale tasks on standard architectures yields alternative benefits, including interpretable results and potential generality of conclusions.
>
> Regarding the impact of the theory beyond what we showed, we want to bring to attention a recent work studying traveling waves in RNNs that found the operator for $T_1$ independently (https://arxiv.org/pdf/2309.08045.pdf). Our work accommodates more general cases that $T_1$.
>
> “a novel perspective on Recurrent Neural Networks (RNNs), framing them as dy- namical systems performing sequence memory retrieval”
>
> Previously, RNNs were used to study memory, but to our knowledge, memory models were not used to explore the behavior of RNNs, and we showed in the paper that this view has benefits.
>
> We showed that RNNs are discrete-time analogs of a sequence memory model GSEMM (a newer class of Hopfield Networks with precisely defined memories and inter-memory interactions). To our knowledge, this is the first mathematically rigorous connection between RNNs and neurocomputational memory models. In this paper, this connection enabled reinterpreting the learned weight matrix of RNNs as stored memories and their interactions. Future work can flesh out the relationship between RNNs and the energy function in memory models.

---

> ### Author Response · Authors · 2023-11-21
>
> “The notion of storing different items in different linear subspaces has also been explored”
>
> We thank the reviewer for the relevant literature. Empirical works have investigated the linear subspaces we presented, mainly from the neuroscience community. To our knowledge, there is no work formalizing these linear spaces, how they interact, and how the two (space and interactions) solve the task at hand. Using the new formalism, we made theoretical predictions on what class of operators could solve the tasks and validated them with experimental results. Note that our theory goes beyond the task T_1 that is analogous to the delayed sequence reproduction task considered in the linked paper on sequence working memory.
>
> “This is obviously not the first time people have studied how neurons process information in RNNs (see the work of Omri Barak and David Sussillo, e.g.).”
>
> Our work builds on the Jacobian spectrum analysis approach of Omri Barak and David Sussilo. This connection is discussed in the Related Works section.  To our knowledge, Jacobian spectrum research has not explicitly delved into the phenomenon of variable binding and defined a notion of storing variables in subspaces of the hidden state.
>
>  Further, the tasks used in the previous studies using spectral interpretations (3-bit flip flop, Frequency-cued sine wave, Context-dependent integration) are intentionally low-dimensional, making it easy to plot and draw conclusions. The variable binding tasks in our paper exhibit very high-dimensional behaviors with complicated spectrum distribution (Figure 5 in the appendix), which means that spectral analysis will not yield easily understandable results. We showed in the paper that all these complicated spectral distributions have a straightforward interpretation with variable memories and their interactions albeit restricted to the variable binding tasks in Section 4. We have made changes to clarify these points in the manuscript.
>
> "What are the tasks? "
>
> We have amended the document with the matrix representation of the composition function for these tasks with a figure above Table 1. They all differ only by the composition function $f$ acting on all the variables.
>
> "I thought u(t) was the input vector, which is 0 for t>s, yet Eqn 2 shows the evolution of u(t) for t>s."
>
> Thank you for pointing this out. Corrected: Eqn 2 is supposed to be y(t) – the network output rather than u(t).
>
> "Why are they focusing on mechanistic interpretability for such a limited behavior? As I understand it, MI can be used to explain any behavior of a neural network."
>
> There are different levels of explainability that MI is used for. Sussilo proposed a very general method, which can be applied to any behavior of neural networks. However, this approach makes interpreting the variable binding tasks in our work challenging. Specifically, the high dimensional nature of the variable binding tasks meant no straightforward interpretation of the Jacobian spectrum.
>
> Our method is specific to the class of variable problems described in Section 4. It provides a better understanding of the RNN behavior beyond what the Jacobian spectrum allows in these tasks. With this specificity, we better understand RNN behavior on these tasks but lose some generality to other tasks. There is a clear gap between what the general method enables in these tasks that the current approach fills. Although the tasks we consider are still simple, repeat copy has been used to test the memory capabilities of recurrent neural networks [1] and more recently found in connection with traveling waves in RNNs [2].
>
> We have clarified these points in the updated paper.
>
> “On the whole I also don't see the specific value in claiming that this analysis is related to episodic memory. Sequential memory, yes. But there is nothing specifically episodic about the motivation for the analyses and the tasks represent serial working memory.”
>
> This analysis is episodic in the sense that it uses an episodic memory model from computational neuroscience and the formalisms (stored memories, inter-memory interactions) it provides to perform the analysis. The definition is in line with GSEMM, and episodic memory as described in neuroscience [3] - which describes the ability of neural networks to store and process temporal and contiguous memory sequences.
>
> [1] Alex Graves, Greg Wayne, and Ivo Danihelka. Neural turing machines. ArXiv, abs/1410.5401, 2014
>
> [2] Thomas Anderson Keller, Lyle E. Muller, Terrence J. Sejnowski, and Max Welling. Traveling waves
> encode the recent past and enhance sequence learning. ArXiv, abs/2309.08045, 2023. URL https://api.semanticscholar.org/CorpusID:262013982
>
> [3] Umbach, Gray et al. “Time cells in the human hippocampus and entorhinal cortex support episodic memory.” Proceedings of the National Academy of Sciences of the United States of America 117 (2020): 28463 - 28474.

---

> > ### Comment · Reviewer_xCsT · 2023-11-22
> >
> > I thank the authors for their additional clarifications. Unfortunately I still think the paper needs work before it can be of use to an interdisciplinary audience.
> >
> > Some specific responses:
> > "Variable binding is a very diverse phenomenon, and tackling all of its aspects in a single paper is intractable. Our class of tasks is a simple restriction that can be used to probe some mechanisms of variable binding."
> > If this is the case then the paper should be much clearer about it and mention the specific features it is trying to probe as the main contribution.
> >
> > "Previously, RNNs were used to study memory, but to our knowledge, memory models were not used to explore the behavior of RNNs"
> > I don't understand this distinction. RNNs are used as memory models. So interpreting them as memory models is core to many previous studies.
> >
> >  "RNNs are discrete-time analogs of a sequence memory model GSEMM (a newer class of Hopfield Networks with precisely defined memories and inter-memory interactions). To our knowledge, this is the first mathematically rigorous connection between RNNs and neurocomputational memory models. "
> >  The hopfield network simply is a (binary) RNN. So there has always been a connection between RNNs and neurocomputational memory models. Therefore this framing of the contribution makes it hard to see what this specific work is achieving.
> >
> > Regarding mechanistic interpretability, I understand that the authors are doing something specific in this paper, but the paper makes more general claims about MI (listed above) that aren't consistent with how the term is used in the literature.
> >
> > Episodic memory has a specific meaning in neuroscience/psychology in that it is autobiographical. In these simple models there is no ability to model specifically autobiographical memory. I understand that the previous literature used this term for sequence memory, but it is best to not propagate that error.

---

### Official Review · Reviewer_Ub1M · 2023-10-30

**Soundness:** 1 poor
**Presentation:** 1 poor
**Contribution:** 3 good
**Rating:** 5
**Confidence:** 4

**Summary:**

This work proposes a mathematical formulation on linear (and potentially non-linear) RNNs to analytically track the storage of all relevant memories in a human interpretable manner. The authors frame this as "Episodic Memory Theory" or EMT. The authors propose a variable binding mechanism, and found that when training RNNs on repeat-copy and compose-copy tasks, the solutions exhibit the theoretically predicted behavior.

**Strengths:**

The mathematical formulation on interpreting internally-stored variables in RNNs is novel and insightful. This work provides an important new tool for the mechanistic deconstruction of RNNs. The most impressive result of the paper is that trained RNNs converge to some intended mechanism, suggesting that the mechanism the authors have found is very likely the most optimal solution for the cost function.

**Weaknesses:**

I have several major issues with this work, as detailed below.

(1) While the mathematical formulation is meaningful, I believe this is not episodic memory (and this proposed method is not a theory of episodic memory). The model and mathematical framework may be inspired by episodic memory models, but the task, objective and entire narrative is largely unrelated to episodic memory. RNNs receiving variable inputs, performing computations based on those variable inputs, and subsequently producing an output is straightforward decision-making or information-processing in many cases, including this work. This is further supported by Figure 1, where performing an addition operation is merely a simple computation that does not even require any memory storage or retrieval. Knowing how to add is not episodic memory. I am open to discussion if the authors still feel it is correctly defined.

(2) The way the entire paper is structured is unnecessarily confusing. I can summarize the work in the following manner:
- Consider an input vector with $d$ dimensions that spans $s$ timesteps, with a total of $sd$ input elements
- The number of neurons in the RNN needs to be greater than $sd$ otherwise superposition effects will occur
- The authors perform a change of basis such that the first $sd$ elements of the latent activity within the RNN now represent the (human interpretable, one-to-one mapped) input sequence with some shifting. This is referred to as "variable memory" by the authors (which is well-named and easily understood if not convoluted by the hard-to-parse narrative leading to its definition).

Right now a reader requires knowledge in RNNs and their applications in neuroscience, as well as some experience in mathematical tools commonly found in modern physics to fully understand the work, when in reality this paper could be written to a pure RNN audience without the Dirac and Einstein notations (at least in the main text), by simply stating that information about the latent is being carefully tracked by a basis transformation and formulating the equations in that context.

(3) The effects described in Figure 3 and 4A are specific to the repeat-copy task. More complex tasks (especially those with non-periodic solutions) may result in weight matrices that are not interpretable or offer any additional insight. Similarly, the compose-copy task, which is surprisingly never defined in this paper (but can be inferred from the Figure 4B to be generating the sequence one unit at a time), is the only task that will give rise to such interpretable values. In general, the authors summarize this class of tasks as $f$ in equation (2). The intended narrative is that the authors are using a class of tasks to elegantly highlight the feature of the proposed method, but my impression is that the method will not be producing anything meaningful beyond this class of tasks.

**Questions:**

See weaknesses.

---

> ### Author Response · Authors · 2023-11-21
>
> We thank the reviewer for deeper insights and for constructive feedback to improve the clarity of the work.
>
> (1)	The definition of Episodic Memory is stipulative: that is, we took an episodic memory model (GSEMM) and showed that RNNs are equivalent to its discretization. Here, episodic memory is defined as the ability of a neural system to store episodes (or distinct sequences of memories). In the context of the paper, we showed it was useful to interpret RNNs using this notion of episodic memory from the memory modeling literature. There are, of course, other definitions of episodic memory. For instance, episodic memory is defined originally as the subjective recall of personal experiences, focusing on the human experience rather than the underlying neural network.  Given the variety of definitions for episodic memory, we don’t think it is fruitful to formalize a single definition encapsulating all aspects of episodic memory. Further, such a precise definition is not necessary for the technical contributions of the paper. We take the point that the addition example in Fig 1 is misleading. We have replaced that example with the generation of the Fibonacci series, which necessitates an explicit variable memory store.
>
> (2)	Thank you for the comments to improve the clarity, we have modified our document clarifying the points of confusion and removing the notation. The intention was that the model of variable binding is defined in the more general vector space – the Hilbert space. We don’t assume any geometry (the definition of the inner product) on the neural representations to formulate the theory. The theory will work irrespective of the geometry of the neural networks representing the solution (Appendix A.3). It so happens that the tasks and RNNs we considered in the paper all represent the solution in a space where the inner product is the vector inner product, and hence, linear algebra can be used. When formulating a theory of neural computation, the rich representational diversity of neural networks needs to be considered, and this notation enables that. We accept that the notation is not standard in the general literature. We take the point that the new notation needs to be better motivated. Given that we found only a simple geometry in these tasks, we have deferred introducing it to future work (and removed the notation from the main document in the updated version). Still, given the utility of the notation to form principled theories for how RNNs work, we don’t see why it cannot be used to inform future developments.
>
> (3)	Figure 3, 4A are results of running the approximation algorithm which can have errors (added a section in the Appendix with formal error analysis). The effect described in the paper is present across all the 4 tasks we considered in the paper (Table 1). Since the variable memory algorithm is approximate, sometimes correct bases cannot be obtained by power iteration, which is why only some tasks can be supported by the empirical algorithm. Figures 3, 4 show the result of running the algorithm to approximate the variable memories, which may have errors when applied in specific tasks. We only show 3 for repeatCopy as the variables there are easier to comprehend than the other composition functions.
>
> "The intended narrative is that the authors are using a class of tasks to elegantly highlight the feature of the proposed method, but my impression is that the method will not be producing anything meaningful beyond this class of tasks."
>
> We want to bring into attention a recent work, that found the operator to solve repeat copy (Fig 4A) in connection to the presence of traveling waves in RNNs (https://arxiv.org/pdf/2309.08045.pdf), suggesting a broader applicability of the theory. Starting with small-scale experiments on standard architectures yields alternative benefits, including interpretable results and potential generality of conclusions. Future work will extend it to more general tasks of interest to the machine learning community.
>
> There is always the added caveat in analyzing RNNs that behaviors like chaos and quasi-periodicity cannot be captured analytically. This is a disadvantage for any analysis of RNNs and not unique to our approach. That said, it should still be interesting that RNNs capture the inherent periodicity of the problems they solve in their dynamical behavior without introducing any additional implicit biases.
>
> We added the 4 task functions as a figure above the Table 1. All these tasks differ only in the definition of $f$, the function computing the final variable memory.

---

### Author Response · Authors · 2023-11-22
**Paper modified for clarity**

We thank the reviewers for constructive feedback on improving the manuscript's clarity. We have rewritten our document to clarify the points of confusion raised by the reviewers. These are the changes in the updated manuscript.

Summary: The rewrite was done to improve clarity - primarily the restrictions we place on variable binding and episodic memory for analytic tractability. The theory, experiments, and contributions of the paper are retained. We also clarified that these restrictions generalize existing analytically tractable methods while retaining the benefits of deconstructing the *full* learned behavior of RNN.

- We have clarified the definition of episodic memory used in the paper, and the restriction on variable binding in the Introduction. We have also added the reference on traveling waves that independently used a specific case of our circuit (the case of the circuit with T_1 and 1-D variables) to improve RNNs.

- Related works: we have added what our work contributes to the three fields.

- Fig1: Replaced the addition example with fibonacci which requires explicit storage and processing of variables. We also added a section in the Appendix A.3 which details how the Fibonacci sequence is generated in the circuit.

- RNN as GSEMM: we have clarified how viewing RNNs as GSEMM (an energy based memory model from computational neuroscience) allows interpretation of the RNN weights as storing memories and inter-memory interactions. The high level proof is provided in the section, and the detailed proof in Appendix A.2

- Variable Binding Tasks: We have added our restricted definition of variable binding in this section, and clarified how the restriction contributes to understanding RNNs by generalizing on existing low-dimensional toy setups that are typically used to probe RNN behavior. Added Appendix Figure 5 now shows the high-dimensional nature of the tasks.

- Table 1: Added a figure above with the matrix representation of the linear composition functions.

- We reworked the sections describing the variable binding circuit and the experimental sections. We have clarified our hypotheses of the variable binding circuit and the validation of the hypotheses in Table 1. Further, we have removed all the newly introduced notation and have deferred it for future work.

- We moved the empirical results to a new section called the "practical considerations of EMT" to clarify the circuit of EMT is present in trained RNNs. Still, additional work needs to be done to extract the circuit and influence its behavior in trained RNNs. We added the variable memory approximation algorithm, explicitly calling it approximation rather than computing, to clarify that the power iteration algorithm approximates variable memories. We have performed an analysis of the error in approximation in Appendix A.6

- We have reworked the Discussion section to clarify what our contributions are to understanding RNNs, and the limitations of the current approach in applying to general RNNs.

---

### Meta-Review · Area_Chair_Ry5G · 2023-12-07

**Metareview:**

The paper proposes the Episodic Memory Theory that casts RNNs as discrete-time analogs of the General Sequential Episodic Memory Model.
The AC and most reviewers agree that the paper makes some novel and potentially significant contributions.

While the authors provided valuable feedback clarifying many of the reviewers' points, significant work is still needed for the contributions to be clarified, and to be appreciated and useful to the community (see the final points of Reviewer xCsT).

We strongly encourage the authors to revise this work to address the limitations regarding the class of tasks and analysis  (see remarks on the method inability to generalize beyond $f$ by reviewer Ub1M,  and related remarks from reviewers xCsT, VSBc),

**Justification For Why Not Higher Score:**

The setting considered is limited, a concern shared by all reviewers.
Serious rewrite is needed to make the paper accessible to the community.

**Justification For Why Not Lower Score:**

N/A

---

### Decision · Program_Chairs · 2024-01-16

Reject